

# Comparing space-based to reported carbon monoxide emission estimates for Europe's iron & steel plants.

Gijs Leguijt[1,2], Joannes D. Maasakkers[1], Hugo A.C. Denier van der Gon[2], Arjo J. Segers[2], Tobias Borsdorff[1], Ivar R. van der Velde[1,3], and Ilse Aben[1,3]

[1]SRON Netherlands Institute for Space Research, Leiden, The Netherlands
[2]Department of Air Quality and Emissions Research, Netherlands Organisation for Applied Scientific Research, TNO, Utrecht, The Netherlands
[3]Department of Earth Sciences, Vrije Universiteit Amsterdam, Amsterdam, the Netherlands

**Correspondence:** Gijs Leguijt (g.leguijt@sron.nl)

**Abstract.** We use satellite observations of carbon monoxide (CO) to estimate CO emissions from European integrated iron & steel plants, the continent's highest emitting CO point sources. We perform analytical inversions to estimate emissions from 21 individual plants using observations from the Tropospheric Monitoring Instrument (TROPOMI) for 2019. As prior emissions, we use values reported by the facilities to the European Pollutant Release and Transfer Register (E-PRTR). These reported emissions vary in estimation methodology, including both measurements and calculations. With the Weather Research and Forecasting (WRF) model, we perform an ensemble of simulations with different transport settings to best replicate the observed emission plumes for each day and site. Comparing the inversion-based emission estimates to the E-PRTR reports, nine of the plants agree within uncertainties. For the remaining plants, we generally find lower emission rates than reported. Our posterior emission estimates are well-constrained by the satellite observations (90% of the plants have averaging kernel sensitivities above 0.7) except for a few low-emitting or coastal sites. We find agreement between our inversion results and emissions we estimate using the Cross-Sectional Flux (CSF) method for the seven strongest-emitting plants, building further confidence in the inversion estimates. Finally, for four plants with large year-to-year variability in reported emission rates or large differences between the reported emission rate and our posterior estimate, we extend our analysis to 2020. We find no evidence in either the observed carbon monoxide concentrations or our inversion results for strong changes in emission rates. This demonstrates how satellites can be used to identify potential uncertainties in reported emissions.

## 1 Introduction

Integrated iron & steel plants are the highest-emitting point sources of carbon monoxide (CO) in Europe. CO is of particular interest as it is both an important air pollutant and relevant for the greenhouse gas (GHG) balance of the atmosphere as an indirect GHG (Daniel and Solomon, 1998). It is a precursor of ozone and reacts with the cleaning agent OH, thereby increasing the atmospheric lifetime of methane (Jacob, 1999; Wuebbles and Hayhoe, 2002). As a product of incomplete combustion, the majority of CO in our atmosphere is emitted by anthropogenic sources (like road transport and industry) and fires (Zhong et al., 2017). As these combustion processes also emit $CO_2$, better knowledge of CO can support our understanding of $CO_2$ emissions



(Park et al., 2021; Wu et al., 2022). The importance of air pollution, both for health effects and for better understanding of our atmosphere, is reflected in regulations by the European Union requiring the reporting of emissions of both GHG emissions
and a large number of air pollutants, including CO, at the facility-level (EUR-Lex, 2006). As these reports are an important factor in policy-making, there is a need for verification of these reported emission rates using additional measurements like satellite data. In this study we will use data from the TROPOMI satellite instrument to estimate the emission rate of the 21 highest-CO-emitting European iron & steel plants.

The iron & steel industry has been marked as an important target for de-carbonization, and there has been a push towards
near-zero carbon-emission production of steel (Skoczkowski et al., 2020; Shahabuddin et al., 2023). However, the largest part of planned capacity (increase) still relies on carbon-intensive production and most of the near-zero production projects are currently in testing stages (Higuera and Van Woensel, 2021; Liu et al., 2022). Combined with the continuously increasing global demand in steel, carbon emissions from steel production have roughly doubled from 2000 to 2020 (Bashmakov et al., 2022). Together, the 21 plants considered emit as much CO as Italy, Europe's fourth highest CO-emitter (E-PRTR, 2023;
Denier van der Gon and CoCO2 WP2, 2022).

The large amount of released carbon is a result of using coal as a reductor in steel production (Zang et al., 2023). Integrated iron & steel plants cover all processes from iron-dust to the production of cast steel. The iron-dust contains a lot of oxides, which have to be separated from the iron. However, as the iron-dust is too fine to be processed in the blast furnaces, it is first agglomerated during sintering: using hot air and coke, the particle size increases. Due to the low combustion efficiency of fine
particles under these conditions (Mohammad et al., 2023), a lot of CO is produced, which is vented into the air with other byproducts (Ho et al., 2013). Subsequently, the combination of sinter and coke is fed into the blast furnace where the oxygen splits from the iron and combines with the carbon molecules of the coke. The liquidized iron is then collected at the bottom of the furnace. Although the reduction of the iron results in a lot of CO, the gas is caught at the top of the furnace and used as fuel (Rackley, 2017). The fraction of carbon in the liquid iron is too high to make steel. Therefore, in the Basic Oxygen
Furnace (BOF), oxygen is led through the iron which binds with the carbon to lower the carbon content to levels appropriate for steel production. Like the blast furnace, the BOF produces a lot of CO that is captured for use as fuel (Rackley, 2017). Annual emission rates for the combination of all processes in the plants can be reported using continuous stack-monitoring, extrapolation of discontinuous measurements, or through calculations using emission factors in combination with activity and production numbers (E-PRTR, 2023).

Independently of directly measuring emission rates and/or activity-based calculation, emission rates can also be determined based on the resulting CO enhancements in the atmosphere. Previous work on regional emission quantification and analysis using satellite-based concentration measurements included the use of MOPITT, Sciamachy, and TROPOMI (e.g. Gloudemans et al. (2006); Khlystova et al. (2009); Worden et al. (2010); Girach and Nair (2014); van der Velde et al. (2021)). Additionally, the resolution of TROPOMI, down to 7 x 5.5 km$^2$ (across x along track), has been shown to be sufficiently high to study
individual cities and CO point sources (Tian et al., 2022; Plant et al., 2022; Leguijt et al., 2023; Goudar et al., 2023; Schneising et al., 2024). The coverage of polar-orbiting satellites like TROPOMI allows for consistent investigation of regions all over the world rather than being confined to places with good reporting infrastructure. Even in Europe, this continuous data availability





is important as there are some gaps in the data gathered by the European reporting framework. As an example: Slovakia has not reported emissions beyond the year 2017 following a change in reporting format (E-PRTR, 2023). For locations with a continuous record of emissions, we will demonstrate that the satellite data can be used as an independent verification of the reported emissions. We use CO observations by TROPOMI for 2019 to perform analytical inversions over the largest 21 European point sources of CO to estimate their emission rates and evaluate consistency with reported emissions. In addition, we perform multi-year analyses for sites with large year-to-year differences in reported emission rates and compare our analytical inversions with other satellite-based emission quantification methods.

## 2    Data & Methods

We use TROPOMI carbon monoxide (CO) data in site-specific analytical inversions to estimate annual CO emissions from the 21 largest iron & steel plants in Europe using their emissions as reported to the European Pollutant Release and Transfer Register (E-PRTR) as prior estimates. We will first describe the TROPOMI data product in Section 2.1. Section 2.2 covers the prior emission data and Section 2.3 describes the Weather Research and Forecasting (WRF) forward model. In Section 2.4 to 2.6, we describe the inversion framework and uncertainty estimation. Finally, in Section 2.7, we describe the Cross-Sectional Flux (CSF) method and the concept of wind rotation, which are supplemental methods to analyse emission rates from satellite data.

### 2.1    TROPOMI carbon monoxide data product

The TROPOMI instrument is a spectrometer on the ESA Sentinel-5 precursor satellite which flies in a sun-synchronous orbit with an equatorial overpass at 13:30 local time (Veefkind et al., 2012). Its swath of 2600 km allows for daily global coverage at a resolution down to 7 x 5.5 km$^2$ (across x along track) for CO. We use the CO operational product version 2.2.0 (Landgraf et al., 2021) which employs the shortwave-infrared CO retrieval (SICOR) algorithm to determine the total column CO concentration based on the absorption of reflected sunlight in the shortwave-infrared band (SWIR, 2305-2385 nm) (Borsdorff et al., 2018). The ground-based Total Carbon Column Observing Network (TCCON, (Wunch et al., 2011)) also measures the total column CO concentrations at specific sites by measuring the spectrum of direct sunlight, allowing for validation of the TROPOMI product. TROPOMI shows good agreement with the unscaled TCCON product with a mean difference per station of $2.45 \pm 3.38\%$ (Sha et al., 2021).

We only use observations with sensitivity to concentrations at the surface and therefore exclude observations with a TROPOMI Data Quality Value (QA Value) below 0.7. The remaining data are either cloud-free or contain only low altitude clouds. Due to the low surface albedo of water, cloudless observations over water bodies result in more uncertain estimates of the CO concentration. We therefore remove cloudless pixels (pixels with a QA Value equal to 1) over water. For all iron & steel plants we analyse TROPOMI data for 2019. For four plants, Arcelor Gent (Belgium), Gijon (Spain), Ostrava (Czech Republic) and Provozovna Třinec (Czech Republic), we also analyse 2020 data.



## 2.2 Prior emissions: E-PRTR reporting framework and TNO emission inventory

The European Pollutant Release and Transfer Register (E-PRTR) is the official pollutant reporting framework of the European Union (EU) (E-PRTR, 2023). Industries in EU member states are required to annually report facility-level emissions of air pollutants and greenhouse gases to air, water, and soil (EUR-Lex, 2006). For steel production, the reporting requirement applies to all facilities with a capacity exceeding 2.5 tonnes of steel per hour (EUR-Lex, 2006). We use the reported emission rates for 2019 except for U.S. Steel s.r.o. in Slovakia for which we use the last reported emission rate (2017) instead.

In addition to a reported emission rate, the E-PRTR database contains information on the methods used to determine each specific emission rate as shown in Figure 1. All measured and calculated emissions are obtained conform to either nationally or internationally approved methods. The label 'measured' applies both to continuous and short-term discontinuous measurements of the emission rate. 'Calculated' emission rates are determined through combined knowledge of activity data (fuel use, steel output) and emission factors while 'estimated' emission rates are determined using non-standardized methods that are not 100 based on publicly available references (ICF, 2020). While Donawitz GesmbH (Austria) mentions the use of stack monitors, which continuously measure the emission rate of gases, and Salzgitter Flachstahl (Germany) reports the use of bi-annual measurements, the majority of the plants do not provide information on the specific method of measurement or calculation, which is in line with findings by ICF (2020).



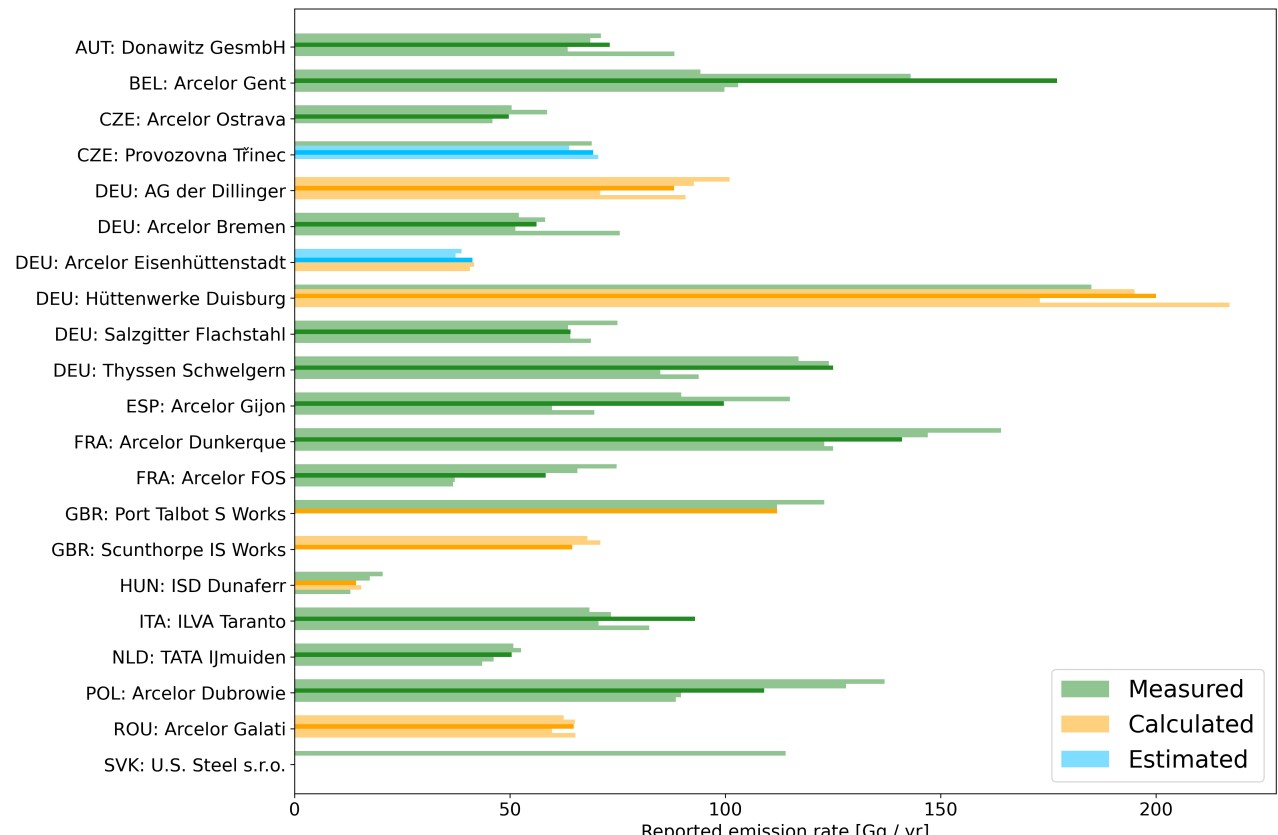

**Figure 1.** Facility-level carbon monoxide emissions as reported to E-PRTR from 2017 (top) to 2021 (bottom). The different methods used to determine these emissions are indicated by the colors, measured (green), calculated (orange), and estimated (blue). The year 2019 (middle), which is used for the analysis in this work is shown more opaquely.

As input to our forward model, we represent anthropogenic CO emissions surrounding the iron & steel plants with the European TNO Greenhouse Gas and co-Emitted species (GHGco) inventory version 4 developed for the EU-horizon CoCO2 project (Kuenen et al., 2022; Denier van der Gon and CoCO2 WP2, 2022). The GHGco inventory focuses specifically on Europe and includes emissions for different source sectors grouped following the Gridded Nomenclature For Reporting (Kuenen et al., 2022). A resolution of $0.05°\text{x}0.1°$ is achieved by combining (1) emission data reported by member states to the Centre on Emission Inventories and Projections of the European Monitoring and Evaluation Programme (EMEP/CEIP), (2) spatial proxies like population density and road networks, and (3) additional datasets like emissions based on reported shipping activity and remotely sensed agricultural fires. The inventory is supplemented with point sources, like iron & steel plants, power plants, and airports, at their exact location. As we use the iron & steel plant emission rate from E-PRTR, we remove the corresponding point sources from the TNO GHGco inventory to avoid double counting of emissions.



## 2.3 Forward model: WRF chemical transport model

We use the Weather Research and Forecasting (WRF) chemical transport model version 4.1 (Powers et al., 2017) to simulate three-dimensional concentration fields around each iron & steel plant for 2019. Table A1 shows the list of the plants we use in the simulations and their locations. All simulations use nested domains centered on the location of the plant with an inner domain (147x147 km$^2$) simulated at 3 km resolution and an outer domain (441x441 km$^2$) at 9 km resolution. Both Thyssen Schwelgern and Hüttenwerke Duisburg (Germany) as well as Arcelor Ostrava and Provozovna Třinec (Czech Republic) lie

within the inner domain of the other, and are combined into one simulation centered midway between the plants.

The E-PRTR emissions for the iron & steel plants are supplemented with anthropogenic emissions from the TNO GHGco inventory (Section 2.2). Both the E-PRTR and TNO emissions are put on a three-dimensional grid using the sector specific vertical profiles provided by Bieser et al. (2011). The temporal profiles applied to the emissions per sector are taken from Guevara et al. (2021). Background concentrations are simulated by using the $0.25°$x$0.25°$ resolution air pollutant forecast

product of the Copernicus Atmosphere Monitoring Service (CAMS) as initial and 6-hourly boundary conditions (Inness et al., 2015). In addition to the iron & steel plant, we simulate each sector in the TNO GHGco inventory in the inner domain separately, as well as the CAMS-based background and enhancements in the inner domain originating from emissions in the four quadrants (NE, SE, SW, NW) of the outer domain.

We model carbon monoxide as an inert gas using the contiguous United States (CONUS) physics suite provided in WRF as

our baseline setup. Over our small model domain, chemical processes have a small impact on the (long-lived) CO enhancements simulated, while the effect of chemistry outside our domain is included in the CAMS boundary conditions. As will be discussed in Section 2.4, it is important for our simulated and observed plumes to have minimal spatial mismatch. However, at the kilometer-scale of TROPOMI observations, exact plumes can be difficult to model. A way to minimize the mismatch is by simulating multiple plumes per day using various model settings (Maasakkers et al., 2022a). Therefore, we perform eight simu-

lations for each location using four different planetary boundary layer (PBL) schemes and corresponding surface layer physics and two different driving meteorological fields. The different planetary boundary layer schemes (Mellor-Yamada-Janjic (MYJ) TKE, YSU, eddy-diffusivity mass flux quasi normal scale elimination (EMF-QNSE), and MYNN 2.5 level TKE scheme) allow for differences in vertical distribution and dispersion speed. As driving meteorological fields, we use the National Centre for Environmental Prediction (NCEP, 2000) and the fifth generation European Centre for Medium-Range Weather Forecasts

(ECMWF) reanalysis products (ERA5) (Hersbach et al., 2020).

To be able to directly compare the simulation to the TROPOMI observations, all simulation output is sampled at the TROPOMI overpass matching footprints of the TROPOMI pixels. The three-dimensional simulation output is converted to a total column by applying the TROPOMI averaging kernel (Landgraf et al., 2021).

## 2.4 Inversion framework

We use an analytical inversion to estimate posterior emissions as described in Brasseur and Jacob (2017).



We optimize the cost function $J(x)$ which is defined as the sum of two parts

$$J(x) = (x - x_A)^T S_A^{-1} (x - x_A) + \gamma (y - Kx)^T S_O^{-1} (y - Kx). \tag{1}$$

The first part defines a penalty on a departure of the state vector ($x$) from the prior inventory emission rates ($x_A$), weighted by the prior error covariance matrix ($S_A$). The second term defines a penalty on the difference between observed ($y$) and simulated ($Kx$) concentrations, weighted by the observational error covariance matrix $S_O$. Here $K$ is the Jacobian matrix of the simulation model with respect to changes in emissions. The regularization parameter ($\gamma$) is used to avoid overfitting to the TROPOMI observations, its determination is discussed in Section 2.5. The optimal posterior solution $\hat{x}$ which minimizes the cost function is given by

$$\hat{x} = x_A + G (y - Kx_A). \tag{2}$$

Here, $G$ is the gain matrix defined as

$$G = \gamma \hat{S} K^T S_O^{-1}, \tag{3}$$

with $\hat{S}$ the posterior error covariance matrix

$$\hat{S} = \left( \gamma K^T S_O^{-1} K + S_A^{-1} \right)^{-1}. \tag{4}$$

With $\hat{S}$ we can calculate the averaging kernel, which gives the sensitivity of the posterior estimate to the true state

$$A = \frac{\partial \hat{x}}{\partial x} = I - \hat{S} S_A^{-1}, \tag{5}$$

where $I$ represents the identity matrix.

To construct $S_A$, we assume a diagonal shape and an uncertainty of $20\%$ for the TNO GHGco inventory. We choose an uncertainty of $10\%$ for the CAMS background following e.g. Maasakkers et al. (2022b), and a $50\%$ uncertainty on the 4 elements adjusting for inflow from the outer domain reflecting the high uncertainties associated with long-range transport. To allow for enough flexibility in the inversion, we use an uncertainty of $30\%$ for emissions from the E-PRTR inventory. However, we test the effect of higher and lower uncertainties in our uncertainty calculation (Section 2.6). To construct the observational error covariance matrix $S_O$, we take the standard deviation of the difference between the simulated concentrations sampled to the TROPOMI footprints and the observations as in Maasakkers et al. (2022a).

As the term $(y - Kx)$ is evaluated for each observation, small mismatches in the exact location of the plume between TROPOMI and the simulation will result in underestimated emissions. This effect can be countered by aggregating the observation to a coarser resolution, in which the simulation and TROPOMI do agree on the position of the plume (Naus et al., 2023). We therefore aggregate TROPOMI observations on a $0.1°$ grid in our inverse analysis, treating each observation as an independent measurement.

Although aggregation reduces the effect of spatial mismatches between simulation and observation, it is not fully eliminated. Following Maasakkers et al. (2022a), the effect of spatial mismatches can be further mitigated by creating an ensemble of



spatially different simulated plumes (Section 2.3). For each overpass of TROPOMI, the simulated plume that best matches the observed data is selected for the inversion, further lowering the model-driven spatial mismatch between observation and simulation. We determine which simulation matches the TROPOMI observation best by performing daily inversions with all 8 simulation outputs and selecting the simulation with lowest optimized observational cost (second term in Eq. 1). The different

180    plumes are simulated with two different driving wind fields and four PBL schemes (Section 2.3) and further expanded by also selecting the simulated plumes an hour before and after the TROPOMI overpass as in Pandey et al. (2019). Figure 2 shows the eight spatially different simulated plumes at overpass-time as well as the TROPOMI plume observed on the same day. The 16 simulated plumes corresponding to the hour before and after the TROPOMI overpass are not shown. Panel I of Figure 2 shows the lowest optimized (posterior) observational cost, and this configuration will therefore be used for the optimization for

185    this day. To further limit the contribution of concentration-mismatches we remove days which have the 20% highest optimized observational cost normalized by the number of pixels. The daily inversions are only used for selection of the best simulation on each day. Afterwards, the best daily simulations are combined into an inversion spanning the full year to determine annual emission rate estimates.

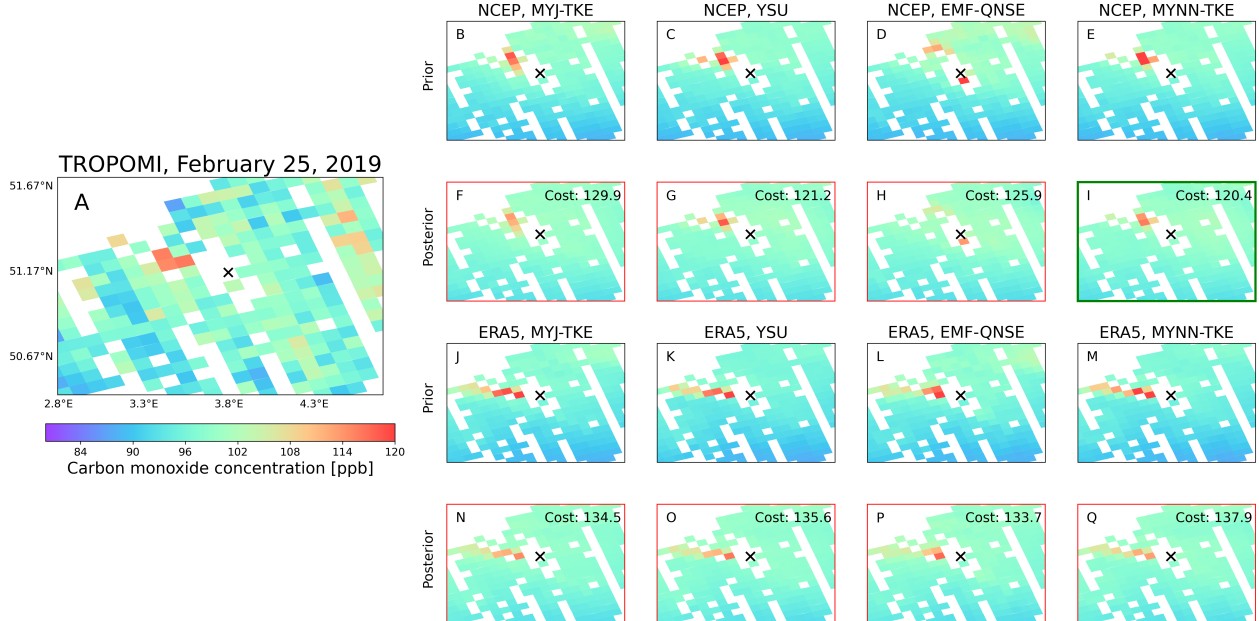

**Figure 2.** Panel A shows concentration as measured by TROPOMI over Arcelor Gent in Belgium (indicated by the x) on February 25th 2019. Panel B-E and J-M show different prior simulations using NCEP and ERA5 meteorological data respectively. The variation between the different NCEP/ERA5 simulations is caused by different planetary boundary layer schemes and surface layer physics (as indicated in the titles). Panel F-I and N-Q show the corresponding posterior concentrations. Out of these posterior simulations, panel I shows the lowest observational cost, making it the best simulation for this particular day.





Figure 2 shows a strong south-west to north-east gradient in all prior simulated concentration fields (panel 2B-E and 2J-M) which is not observed in the TROPOMI data. Such strong gradients are not often observed in the simulated data, but days that do have them will negatively impact the accuracy of the inversion result as the over- and underestimates in the simulated concentrations will be compensated for by respectively lowering or increasing the emission rates over affected areas. To reduce the impact of mismatches between the simulated and observed background, we allow our inversion to optimize the background at daily rather than yearly frequency to prevent biases from aliasing into the emissions estimate. We further split the background into a mean background (which is uniform for all pixels per day) and a deviation from the mean (which is any remaining spatial pattern present in the CAMS-based simulated background). These two parts of the background are added individually to the state vector, yielding two state vector elements per overpass of TROPOMI and giving additional flexibility to the inversion. Panel 2F-I and 2N-O show this flexibility results in a reduced spatial gradient in the posterior simulations, better matching the TROPOMI observation.

## 2.5 Regularization parameter

Because of the large number of TROPOMI observations and the assumption of a diagonal observational error covariance matrix, there is a risk of overfitting to the observations. We therefore apply regularization parameter $\gamma$ to the observational part of the cost function (Equation 1). To determine an appropriate value for $\gamma$, we use the L-curve criterion as described in Hansen (1999). As we are mostly interested in correctly quantifying emissions from the iron & steel plant, we reduce the background contribution to the cost-function by scaling the background by the mean difference between simulation and observation over the full year before determining $\gamma$. The resulting L-curve can be found in Appendix B, from which we conclude $\gamma = 0.1$ is appropriate.

## 2.6 Uncertainty analysis

To evaluate the uncertainty of our posterior emission estimates, we perform an ensemble, varying the relevant parameters in our inversion framework. We report the full spread of this ensemble for each plant as uncertainty. The range over which each parameter was varied can be found in Appendix C. Figure 3 shows the resulting spread in emission rates related to each varied parameter. Not optimizing the background daily and different aggregation resolutions result in large spreads, exceeding those resulting from the use of different wind products and choices in data filtering. As an alternative to using the observational cost function for selecting the best matching simulation (Section 2.4), we select the simulation based on the highest posterior scaling. We include this (potentially high-biased) approach within the uncertainty ensemble to get a conservative uncertainty range. Although the choice of the regularization parameter has a small effect on the emission estimate for most plants, it affects a few individual sites more heavily than any of the other variables because they have relatively few observations, and the lower $\gamma$ values then keep the estimates close to the prior. The posterior estimate is relatively insensitive to variation of the prior, showing that the emission estimates are strongly determined by the TROPOMI observations.





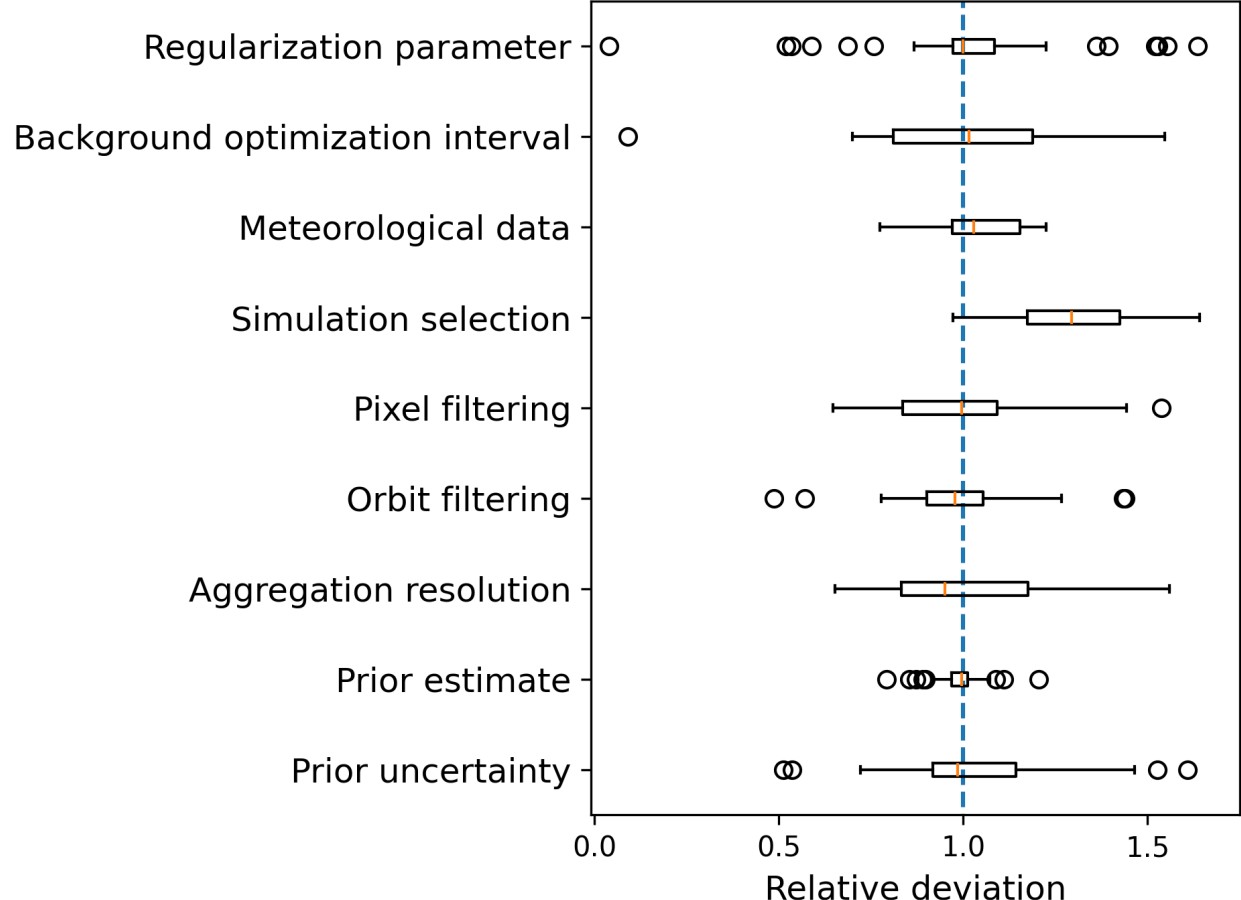

**Figure 3.** The uncertainty related to each parameter for all plants. The x-axis shows the relative deviation from the base posterior estimates. The boxplots show the collection of all investigated plants with the x-axis showing the resulting emission rates normalized by their base posterior estimate. Each boxplot consists of the estimates for each plant for the entire spread in the variable as classified in Appendix C.

## 2.7 Additional quantification methods

We compare and supplement our inversion approach with two additional methods; the Cross-Sectional Flux (CSF) method, and an approach based on oversampling and wind rotation. Both methods solely rely on the CO-concentrations measured by TROPOMI and a wind field, without incorporating any prior knowledge on emission rates or using simulations of atmospheric concentrations.

The CSF method (Varon et al., 2018) is a 'mass balance' emission quantification method that calculates the particle flux at different distances from a source. First, CO enhancements are integrated over cross-sections perpendicular to the plume. Multiplied by the wind speed, the full integral over each cross-section gives an emission rate estimate. By repeating this





procedure at different distances from the source, an average emission rate corresponding with the observed plume is calculated. The simplicity of the method allows for fast application to different locations at the cost of a larger uncertainty and higher
minimum emission threshold than methods relying on large atmospheric transport models. We perform the CSF as in Leguijt et al. (2023) using 10-meter altitude winds from ERA5 (Hersbach et al., 2020). As uncertainty we report the full range of the same ensemble members as used in Leguijt et al. (2023).

To investigate whether year-to-year variation in inversion-based emission rate estimates are consistent with trends in observed CO concentrations, we also perform a method based on an oversampled wind rotation as in Clarisse et al. (2019).
Because of variation in the wind direction, plumes at different days will point in different directions and oversampling measured concentrations without taking wind-information into account will result in a diffuse enhancement. Valin et al. (2013) and Pommier et al. (2013) showed that the spatially averaged concentrations retain a plume-like shape if the enhancements are rotated around the source location such that the wind points in the same direction. Using the approach as developed in Maasakkers et al. (2022b), we oversample wind rotated concentration fields and use these as an indication of emission trends
rather than a determination of absolute emission rates.

## 3    Results & Discussion

We first discuss the performance of our inversion in Section 3.1, followed by a comparison of the satellite-based emission estimates with reported emission rates in Section 3.2. To explain any differences between the two, we have extended our analysis to 2020 for some of the investigated plants for which the results are shown in Section 3.4. Section 3.3 explores
consistency with the model-independent Cross-Sectional Flux (CSF) method.

### 3.1    Inversion performance

In Figure 4A we show the difference between the prior simulation and observations for 2019 over Arcelor Gent (Belgium), gridded at $0.05°$. Throughout the domain, excluding five pixels at the center, the concentrations measured by TROPOMI exceed the simulated concentrations. Figure 4C shows the corresponding difference plot for the posterior simulation after
optimization of the state vector. This figure shows a reduced bias (1.89 to 0.01 ppb) and absolute bias (2.00 to 0.95 ppb), and a higher correlation (0.80 to 0.83) between simulation and observation, as expected from the optimization. In addition, no spatial patterns are visible in the resulting difference map. The difference between prior and observation can largely be explained by differences in simulated background. Figure 4B shows the difference between simulation and observation where only the background has been optimized. Although the largest part of the domain shows better agreement to the observations than the
prior simulation (mean bias: -0.11, mean absolute bias: 0.97, correlation: 0.83), the simulated concentrations above the iron & steel plants show significant differences. This shows the inversion framework is sensitive to emissions from the iron & steel plant specifically. The corresponding values for the other plants are shown in Appendix D.





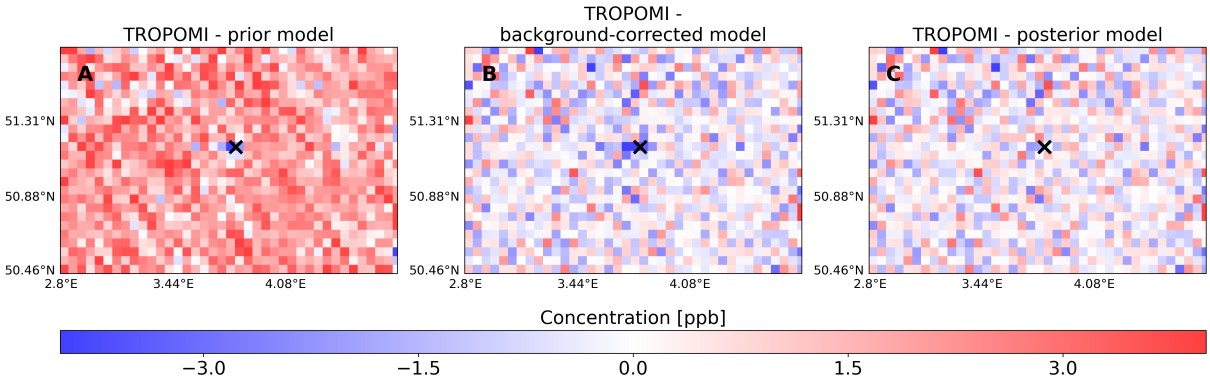

**Figure 4.** Panels A-C show the difference between the prior and posterior simulation and TROPOMI observations over Arcelor Gent (Belgium, marked by the x) for 2019 aggregated at $0.05°$. Panel A shows the prior difference, where TROPOMI observations are higher than the simulated concentrations throughout most of the region. Panel B shows the impact of optimizing the background, showing a smaller difference, except around the iron & steel plant. In the posterior difference shown in Panel C, there is no clear pattern visible in the difference between the simulation and TROPOMI.

## 3.2 TROPOMI emission estimates

Figure 5 shows the TROPOMI-based posterior emission estimates compared to the prior emission rates from E-PRTR. Nine
out of twenty-one posterior emission estimates agree with E-PRTR within their uncertainty range and ten posterior estimates
lie within 20% of the reported values. The TROPOMI estimates show a correlation of 0.86 with the E-PRTR dataset and are
on average 17% lower as reflected in Appendix E.

The hashed bars show the diagonal elements of the averaging kernel as given in equation 5, which reflect the ability to
constrain the iron & steel plant emission estimates based on the satellite observations (Jacob et al., 2016). These elements
range between 0 and 1, with 1 indicating the posterior is fully determined by the observations and values close to 0 resulting
in posterior estimates that are mostly determined by the prior. Most of the plants, 67% (90%), have averaging kernel values
above 0.8 (0.7) respectively, owing to the large number of TROPOMI observations. The exception is ISD Dunaferr (Hungary)
with a diagonal element equal to 0.18. This is also the plant with the lowest prior and posterior emission estimate, resulting in
a low sensitivity of the observations to the (small) emissions.

The inversion-based emission estimates for the German plants (Arcelor Bremen, Eisenhüttenstadt, AG der Dillinger, Salzgitter Flachstahl, Thyssen Schwelgern and Hüttenwerke Duisburg) agree within error bars with the emission rates determined by
Schneising et al. (2024) using TROPOMI data in a mass balance approach. However, for these six plants, our emission estimates lie on the lower edge of their uncertainty estimates. Arcelor Gent, Gijon, Dunkerque, Ostrava, and ILVA Taranto show
considerably lower posterior emission estimates even though their reported emissions are based on measurements (as indicated
by the green color). The same applies to U.S. Steel s.r.o (Slovakia), although the emission rate was reported for the year 2017.
This indicates our estimates may be conservative as a perfect estimate requires a sufficient spatial match between the modeled





and observed plume. Additionally, for Arcelor Gent, Gijon and ILVA Taranto reported emissions drop by respectively 42, 40 and 24% for 2020 compared to 2019 (Figure 1). We investigate whether the disagreement between our posterior estimates and E-PRTR is persistent in 2020 in Section 3.4. Port Talbot S Works and Hüttenwerke Duisburg show large uncertainty ranges.

The low values for Talbot originate from the $\gamma = 1$ and yearly-background optimization ensemble members. As Port Talbot is coastal, a lower number of observations and discrepancies between land and water pixels might create difficulties for the inversion framework. A regularization parameter equal to 1 has a risk of overfitting the observational data and, specifically for Talbot results in a close to zero emission rate estimate. Similarly, not allowing for daily optimization of the background could result in differences between land and water pixels being wrongly interpreted as the effect of the plant. However, the other

coastal plants (Arcelor Dunkerque, FOS, Gijon, ILVA Taranto and Tata IJmuiden) do not show larger uncertainties compared to inland locations.

The high posterior estimates in the uncertainty range for Hüttenwerke Duisburg come from the ensemble members with $\gamma = 1$ and prior uncertainty equal to 50%. Both would allow the inversion to wrongly attribute emissions from the neighbouring Thyssen Schwelgern plant (±10km difference) to Hüttenwerke Duisburg. However, the corresponding ensemble members for

Thyssen Schwelgern are only 15-20% lower than the base inversion, meaning the summed emission for the two plants is considerably higher than the base inversion in these ensemble members. The fact that the posterior estimate for specifically Hüttenwerke Duisburg is very uncertain shows the inversion has limited ability to distinguish between two spatially close point sources.



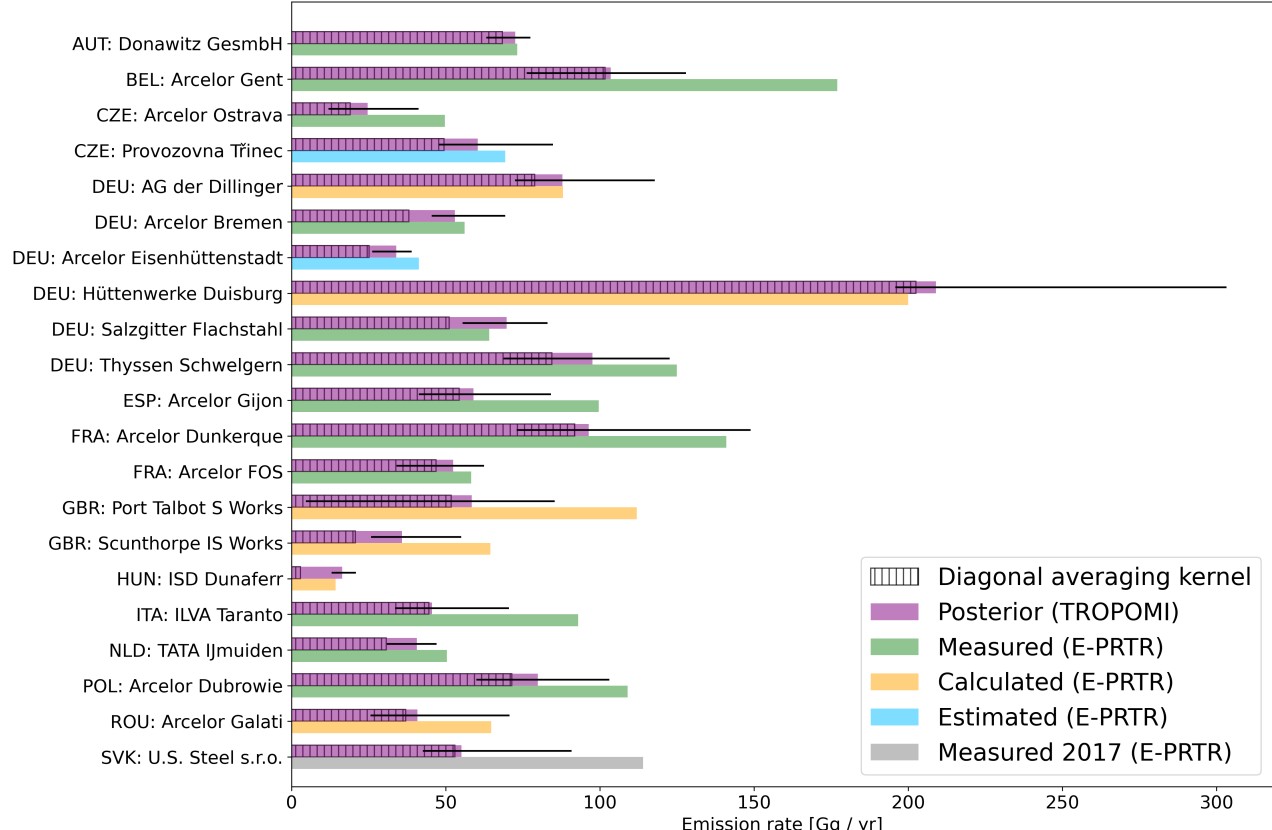

**Figure 5.** Comparison between our posterior emission estimates (purple) and the emissions reported to E-PRTR for 2019. The colors represent the method used to report the emissions to the E-PRTR framework, these are the same as in Figure 1 except for U.S. Steel s.r.o., where the gray bar reflects the difference in year between the TROPOMI estimate and the reported emissions. The error bars on the TROPOMI-based estimates show the full spread of the inversion ensemble. The diagonal elements of the averaging kernel for each steel plant are shown as hashed bars, these elements are a measure for the extent to which the final emission estimates are constrained by the satellite observations. The value of the averaging kernel corresponds to the fraction of the bar that is shaded.

## 3.3 Consistency with Cross-Sectional Flux method

In addition to making use of inversions, TROPOMI CO-data can be used to determine emission rates using simpler "mass balance" methods like the Cross-Sectional Flux (CSF) method. Leguijt et al. (2023) found a 100 Gg per year lower limit for the CSF to be trustworthy when applied to TROPOMI data. Seven of the studied plants; Arcelor Dunkerque, Gent, Dubrowie, Port Talbot S Works, Thyssen Schwelgern, Hüttenwerke Duisburg, and U.S. Steel s.r.o., have reported emission rates above 100 Gg per year and therefore merit the application of the CSF.

Figure 6 shows a comparison between the two methods for the applicable plants. All estimates agree within uncertainty. The CSF estimates of four out of seven plants are higher than the inversion estimates which could support the notion that





the inversion estimates are conservative. Thyssen Schwelgern and Hüttenwerke Duisburg lie only 10 km from one another, meaning the enhancements resulting from emissions at these locations can overlap. The CSF assumes singular point-like sources and is therefore not fully applicable to this situation. This can explain the disagreement between both methods at

Thyssen Schwelgern. Although the reported emission rates for Port Talbot S Works and U.S. Steel s.r.o. are above the 100 Gg per year emission threshold for the CSF, the inversion-based estimates of 58 and 55 Gg per year fall considerably below this value. When applying the CSF to these locations, we find estimates of 59 and 34 Gg per year, which lie well below the 100 Gg per year threshold. Therefore, while we have limited confidence in the CSF retrieved emission rates, they do support that the emission rates could be lower than reported.

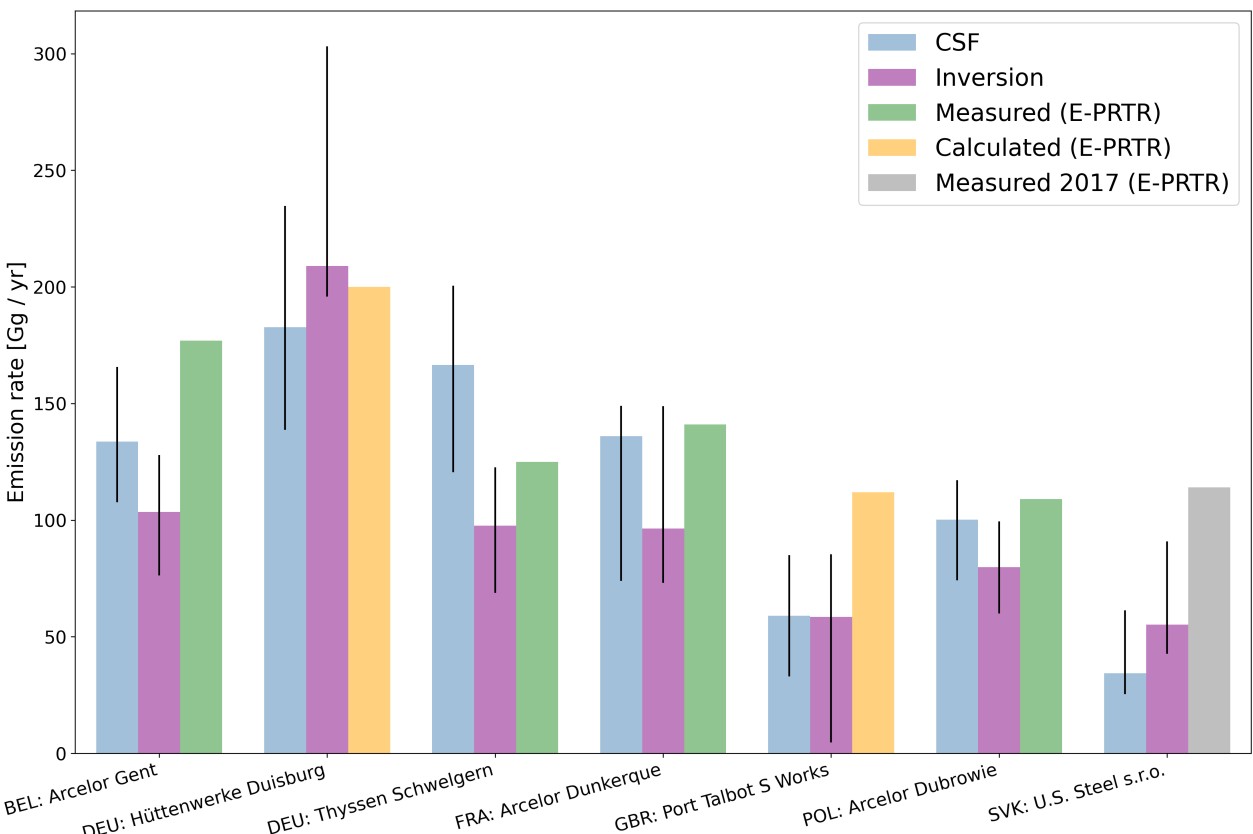

**Figure 6.** Comparison of inversion-based TROPOMI emission estimates with estimates from the mass-balance Cross-Sectional Flux (CSF) method. Emissions reported to E-PRTR are also shown, using the same colors as in Figure 5 to represent the reporting method. As the lower limit for the TROPOMI-based CSF method was estimated at 100 Gg per year, only the plants with prior or posterior estimates above this value are compared.





## 3.4 2020 analysis

As discussed in Section 3.2, we extend our analysis to 2020 for four of the plants showing considerable differences between prior and posterior for 2019: Arcelor Gent, Gijon, Ostrava and Provozovna Třinec. The latter two show little variation in reported emissions from 2019 to 2020 (Figure 7). On the other hand, both Arcelor Gent and Gijon show a sharp drop in reported emissions (42 and 40% respectively) from 2019 to 2020, with the 2019 Arcelor Gent reported emission exceeding the mean of the surrounding four years by 61%. Figure 7 also shows the inversion results for these four plants for 2019 and 2020. Arcelor Ostrava shows little difference between the 2019 and 2020 posterior estimates, as expected from the limited variation in reported emission rates. The inversion-based emission rate estimate for Provozovna Třinec increases from 60 (48-91) to 87 (77-102) Gg/yr, despite having no variation in reported emissions. Both Arcelor Gent and Gijon, which have a very different prior for the 2019 simulations than for the 2020 simulations, show much less variation in the posterior estimate than in the reported emissions. The prior emission estimates for 2020 actually agrees better with the posterior 2019 estimates for both plants. The 18% reduction in the posterior estimate for Arcelor Gent also lies within the uncertainty range of our estimate, showing no clear indication of a reduction in emission from 2019 to 2020 contrary to what is suggested by the large difference in reported emissions for those years. Arcelor Gijon shows a 22% increase in posterior emission rate as opposed to a decline, although this increase lies within the uncertainty range of the 2019 estimate.



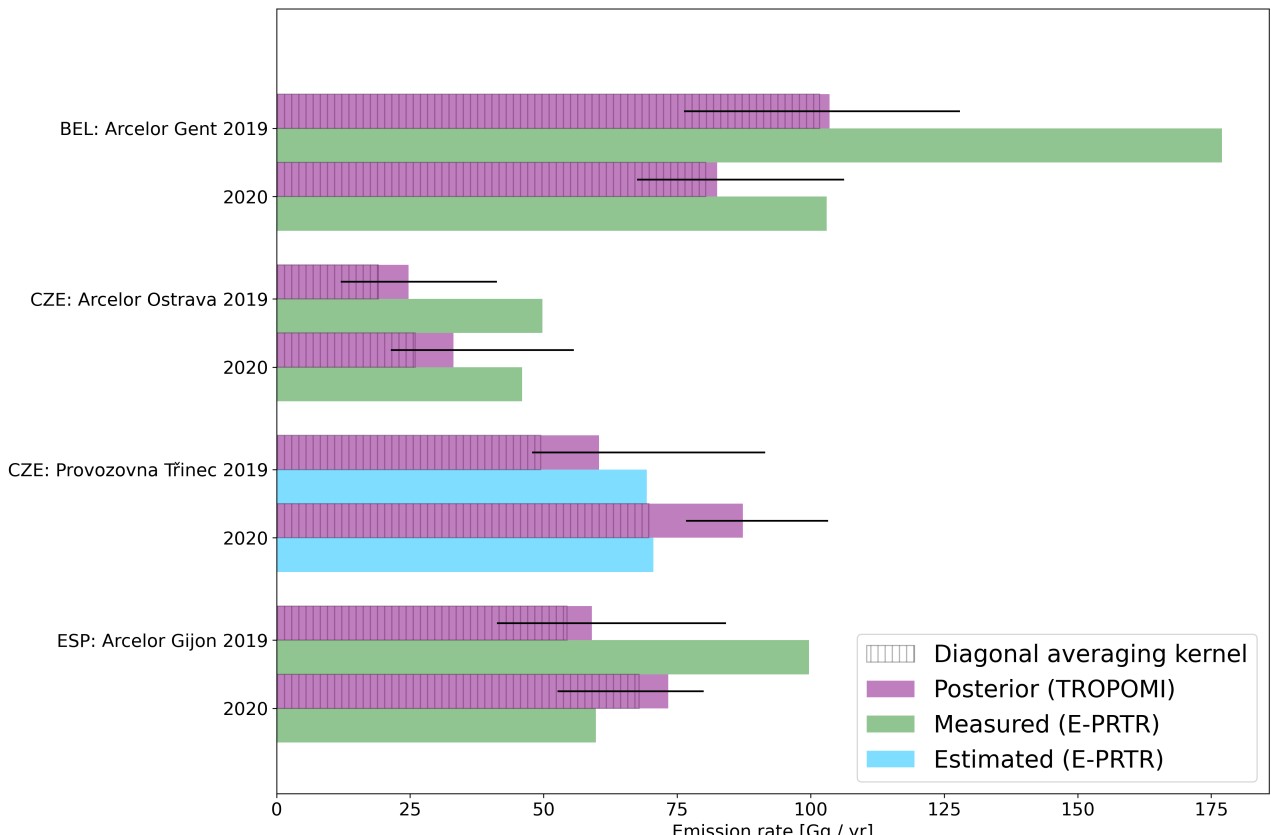

**Figure 7.** Extended inversion analysis for four plants using 2020 TROPOMI data. Each plant shows 4 bars, including the reported emission as well as the inversion emission estimate for the years 2019 and 2020. As in Figure 5, the colors of the reported emissions represent how they were derived and the hashed bars show the diagonal averaging kernel values.

To verify the lack of variation observed in our posterior estimates, we perform annual wind rotations (Section 2.7). Figure 8 shows the results for the four plants for 2019 and 2020. We estimate the enhancements related to the plant as the difference between $0.1°$ x $0.2°$ (width x length) boxes downwind and upwind of the plant. To quantify uncertainties on year-to-year comparisons, we vary the dimensions of up- and downwind boxes simultaneously by up to 30%, and report the full spread. Due to its coastal location, the wind rotations over Arcelor Gijon do not converge in a clear plume and cannot be used to estimate a variation in enhancement. Over Arcelor Gent, we find an enhancement of 2.1 (2.0 - 2.3) ppb for 2019 and 2.0 (1.9 - 2.1) ppb for 2020. This 5% (4% - 11%) decrease in enhancement is more in line with the 20% (3% - 37%) reduction in our posteriors than with the 42% reduction in reported emissions. For Arcelor Ostrava, we find a decrease in enhancement of 22% (10% - 27%). This decrease may partly be attributed to misalignment of the plume and wind-direction, which is more prominent in 2020. Over Provozovna Třinec, we see a 15% (0% - 19%) increase in enhancements, which agrees well with the



reported lack of variation between 2019 and 2020. For both Arcelor Ostrava and Provozovna Třinec the percentage changes in

wind-rotated enhancements are consistent within error bars with the year-to-year variations in our posterior estimates.

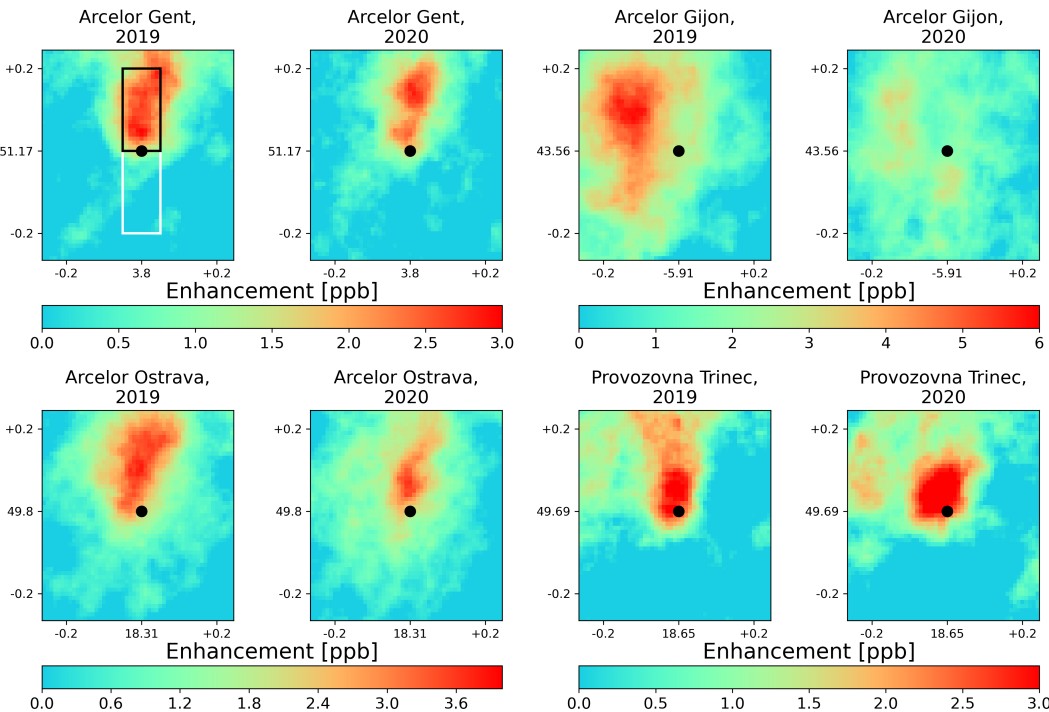

**Figure 8.** TROPOMI wind-rotated averaged concentrations over Arcelor Gent, Gijon, Ostrava, and Provozovna Třinec for 2019 and 2020 oversampled to 0.01 degree. The mean CO concentration across the scene has been subtracted from each image to be able to compare the different years. The first panel shows the boxes used to calculate the upwind (white) and downwind (black) concentrations.

## 4  Conclusions

We performed analytical inversions with 2019 TROPOMI satellite data to determine annual carbon monoxide emission rates for 21 European integrated iron & steel plants. These plants are the highest emitting CO point-sources in Europe. Their

reported facility-level emissions to E-PRTR are used as prior estimates in our inversions. Per site, the inversion uses one of 8 simulations with different meteorology for each day to reach optimal spatial agreement between observation and simulation. We allow further freedom in the inversion by optimizing the CAMS-based background on a daily basis, rather than performing an annual scaling.



We find that the posterior estimates for plants with reported emission rates above 50 Gg per year (the majority of the plants), can be constrained with the satellite observations (averaging kernel values above 0.7 for $90\%$ of the plants). Our emission estimates show a high correlation of 0.86 to the emission rates reported to E-PRTR with 9 out of 21 plants agreeing within estimated uncertainties. For the remaining 12 plants, we find lower posterior emission rates than reported to E-PRTR, suggesting our method gives conservative emission estimates. For one of our six coastal sites, and for a location with a nearby (within 10 km) point source, we find large differences in posterior estimates for different inversion set-ups, highlighting that results for these plants and other plants in similar situations should be treated with care.

For the 7 plants reporting emission rates above 100 Gg per year, we perform an additional emission quantification using the CSF method which does not rely on prior emission information. We find that the CSF-based emission estimates agree with the inversion-based estimates for isolated plants. For two plants in the United Kingdom and Slovakia, the CSF estimates fall below the 100 Gg reported as the lower limit for this method to produce reliable results. However, these low estimates do provide additional confidence in our inversion-based estimates that are also significantly lower than the reported emissions.

We expand our analysis to 2020 for four plants that show large 2019 discrepancies. The inversion estimates for 2019 and 2020 agree with each other, showing the robustness of the method. For Gent (Belgium) and Gijon (Spain), the reported emission rates for 2020 are 40% lower than those reported for 2019 while they agree with the 2019 and 2020 inversion estimates, raising questions on the reported emissions for 2019. Comparison of wind-rotated oversampled TROPOMI data for 2019 and 2020 for Gent also shows no indication for a large difference in emission rate between the years. This example shows how these satellite analyses can be used to identify uncertainties in reported emissions. In general, the good agreement between our results and reported emissions indicates that our framework can be used as a measurement-based approach to estimate CO emissions from large steel plants where site-specific measurements are limited or not available.

*Code and data availability.* TROPOMI (https://doi.org/10.5270/S5P-bj3nry0, Copernicus Sentinel- 5P, 2021) are publicly available at https://dataspace.copernicus.eu (last access: 2 April 2024). ERA5 wind data are available via https://www.ecmwf.int/en/forecasts/dataset/ecmwf-reanalysis-v5 (Hersbach et al., 2022). WRF-Chem code is available at https://github.com/wrf-model/WRF/releases (Powers et al., 2023); in this work, version 4.1.5 was used. Open fire emissions from GFAS are available at https://atmosphere.copernicus.eu/global-fire-emissions (Kaiser et al., 2022). Emissions reported to E-PRTR are publicly available at https://industry.eea.europa.eu/download (last access: 2 April 2024). The TNO-GHGco-v4 inventory with point sources at exact location (Kuenen et al., 2022; Denier van der Gon and CoCO2 WP2, 2022) is available upon request from TNO (contact: Hugo Denier van der Gon, hugo.deniervandergon@tno.nl).





## Appendix A: Iron & steel plant locations

| Name | Country | Latitude/longitude | Name | Country | Latitude/longitude |
|------|---------|--------------------|------|---------|--------------------|
| Donawitz GesmbH | Austria | (47.380,15.066) | Arcelor Dunkerque | France | (51.016,2.336) |
| Arcelor Gent | Belgium | (51.582,3.819) | Arcelor FOS | France | (43.466,4.937) |
| Arcelor Ostrava | Czech republic | (49.796,18.306) | Port Talbot S Works | United Kingdom | (51.556,-3.765) |
| Provozovna Třinec | Czech republic | (49.688,18.647) | Scunthorpe IS Works | United Kingdom | (53.581,-0.62) |
| AG der Dillinger | Germany | (49.357,6.754) | ISD Dunaferr | Hungary | (46.943,18.941) |
| Arcelor Bremen | Germany | (53.125,8.687) | ILVA Taranto | Italy | (40.517,17.2) |
| Arcelor Eisenhüttenstadt | Germany | (52.166,14.618) | TATA IJmuiden | Netherlands | (52.477,4.592) |
| Hüttenwerke Duisburg | Germany | (51.368,6.712) | Arcelor Dubrowie | Poland | (50.080, 20.092) |
| Salzgitter Flachstahl | Germany | (52.155,10.403) | Arcelor Galati | Romania | (45.438,27.972) |
| Thyssen Schwelgern | Germany | (51.507,6.735) | U.S. Steel s.r.o. | Slovakia | (48.618,21.198) |
| Arcelor Gijon | Spain | (43.556,-5.911) | | | |

**Table A1.** Location of the investigated iron & steel plants.




## Appendix B: Regularization factor determination

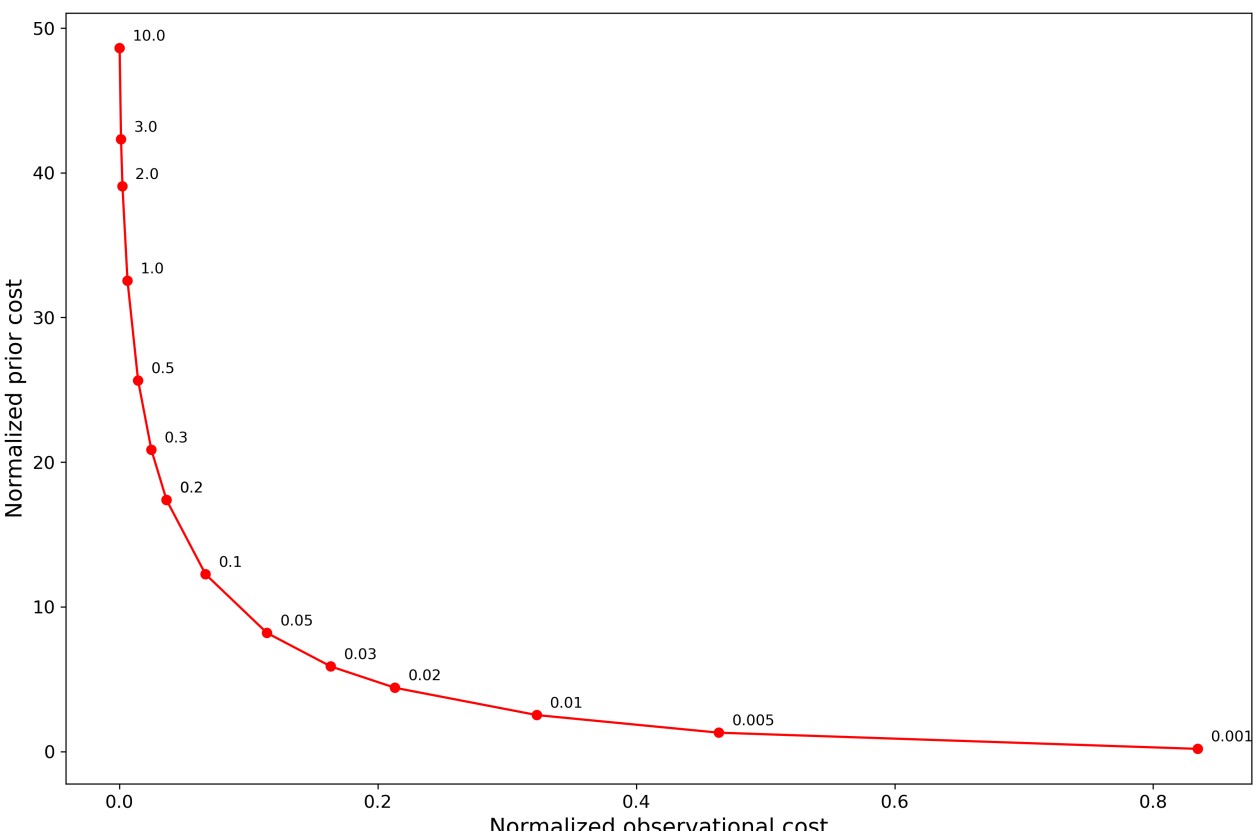

**Figure B1.** Determination of the regularization parameter using the L-curve criterion from Hansen (1999) as discussed in Section 2.4. Different values of the regularization parameter are indicated in the graph with the $y$-axis showing the cost related to deviation from the prior and the $x$-axis showing the observational cost. Both costs have been normalized by the number of state-vector elements and the number of observations respectively. For low gamma values, emission estimates do not deviate from the prior, which results in large differences between simulation and observation. For too high values of gamma, the inversion overfits the observations, resulting in a strong increase in the prior cost. Based on the bend in the L-curve, we chose a value of 0.1 for the regularization parameter.

## Appendix C: Uncertainty in our inversion estimates

To estimate the uncertainty and sensitivity of our inversion-based emission estimates, we perform an ensemble of inversions varying different parameters. Table C1 contains the full list of ensemble members, which we describe in detail here: (1) 0.1 has been established as a suitable value for the regularization parameter $\gamma$ in Appendix B. In the ensemble we include inversions with $\gamma = 0.01$ and $\gamma = 1$. (2) We optimize a daily background optimization in the base inversion, in the ensemble we include inversions that only optimize a single scaling of the background. (3) The ensemble includes inversions that use only NCEP or



ERA5 data. We also add ensemble members that only use the simulations sampled at the TROPOMI-overpass time, instead of

using the hour before and after overpass. (4) Whereas the base inversion uses the optimized observational cost to select the daily

simulation, the ensemble includes inversions which use the maximal posterior scaling for simulation selection. (5) As members

in the ensemble we include inversions that use different pixel- and orbit-filtering than the base inversion. On the pixel-level, we

include inversions with a minimum TROPOMI Data Quality Value (QA Value) of 1.0 compared to 0.7 in the base inversion.

This strict filtering removes up to 87% of data compared to the base inversion. Due to the low number of observations, we use

a regularization value of 0.5 (as opposed to 0.1) for this specific ensemble member, which was determined in the same way as

described in Appendix B. In addition to filtering based on QA Value, we include inversions with a lower maximum allowed

scattering layer height. The base inversion uses a value of 5 km, which corresponds to a QA Value of 0.7. We lower this value

to 0.5 km, which corresponds to QA Value = 1.0. This inversion differs from inversion using the QA Value = 1.0 filtering as the

QA Value imposes additional bounds on the aerosol optical thickness. By not further constraining the optical thickness, about

50% of TROPOMI observations with QA Value of 0.7 and higher are retained. (6) Orbit-filtering in the base inversion is done

by removing the 20% overpasses with the highest optimized observational error per pixel. In the ensemble, we include both

cases where we remove 40% of overpasses, as well as cases in which we retain all overpasses. (7) Within the ensemble, the

resolution to which we aggregate the simulations and TROPOMI observations is increased from $0.1°$ to $0.15°$. We also include

inversions in which no aggregation is applied. (8) Within our ensemble we change the prior by up to 30%. (9) We also vary the

prior uncertainty from the 30% uncertainty used in the base inversion. Our ensemble members include uncertainties of 10, 20,

40 and 50%.



| | Variable | Default value | Range |
|---|---|---|---|
| **Regularization** | | | |
| (1) | $\gamma$ | 0.1 | 0.01 - 1.0 |
| **Background** | | | |
| (2) | Optimisation interval | daily | yearly |
| **Meteorological** | | | |
| (3) | Wind product | NCEP + ERA5, overpass time $\pm$ 1 hour | NCEP overpass |
| | | | ERA5 overpass |
| | | | NCEP + ERA5 overpass |
| | | | NCEP, overpass $\pm$ 1 hour |
| | | | ERA5, overpass $\pm$ 1 hour |
| **Simulation selection** | | | |
| (4) | Metric of selection | minimal observation cost | maximum posterior scaling |
| **Filtering** | | | |
| (5) | Minimum QA Value | 0.7 | 0.7 - 1.0 |
| | Maximum scattering layer height | 5 km | 0.5 km - 5.0 km |
| (6) | Removing worst matching overpasses | 20% | 0% - 40% |
| **Aggregation** | | | |
| (7) | Resolution | 0.1° | no aggregation - 0.15° |
| **Prior estimate** | | | |
| (8) | Scaling | 1.0 | 0.7 - 1.3 |
| **Prior uncertainty** | | | |
| (9) | Prior uncertainty | 30% | 10% - 50% |

**Table C1.** Full range over which variables were varied in the uncertainty ensemble.



## Appendix D: Inversion performance

| Name | Prior model | | | Background corrected model | | | Posterior model | | |
|---|---|---|---|---|---|---|---|---|---|
| | abs. error | mean error | $R^2$ | abs. error | mean error | $R^2$ | abs. error | mean error | $R^2$ |
| Donawitz GesmbH | 2.42 | 2.25 | 0.78 (0.86) | 1.36 | -0.05 | 0.78 (0.89) | 1.36 | -0.05 | 0.78 (0.89) |
| Arcelor Gent | 2.00 | 1.89 | 0.80 (0.81) | 0.97 | -0.11 | 0.83 (0.82) | 0.95 | 0.01 | 0.83 (0.85) |
| Arcelor Ostrava | 2.57 | 2.47 | 0.72 (0.74) | 1.08 | -0.08 | 0.71 (0.75) | 1.07 | -0.02 | 0.71 (0.77) |
| Provozovna Třinec | 2.57 | 2.47 | 0.72 (0.77) | 1.08 | -0.03 | 0.71 (0.86) | 1.07 | -0.02 | 0.71 (0.86) |
| AG der Dillinger | 1.67 | 1.55 | 0.73 (0.79) | 0.97 | 0.01 | 0.75 (0.80) | 0.97 | 0.01 | 0.75 (0.80) |
| Arcelor Bremen | 2.35 | 2.31 | 0.76 (0.82) | 0.92 | 0.00 | 0.79 (0.82) | 0.92 | 0.00 | 0.79 (0.82) |
| Arcelor Eisenhüttenstadt | 2.10 | 2.03 | 0.78 (0.74) | 0.95 | 0.01 | 0.81 (0.75) | 0.95 | 0.01 | 0.81 (0.76) |
| Hüttenwerke Duisburg | 1.56 | 1.31 | 0.79 (0.66) | 1.01 | 0.02 | 0.80 (0.80) | 1.00 | 0.00 | 0.80 (0.82) |
| Salzgitter Flachstahl | 1.66 | 1.51 | 0.76 (0.75) | 0.97 | 0.01 | 0.78 (0.77) | 0.97 | 0.00 | 0.78 (0.78) |
| Thyssen Schwelgern | 1.56 | 1.31 | 0.79 (0.66) | 1.01 | -0.09 | 0.80 (0.80) | 1.00 | 0.00 | 0.80 (0.81) |
| Arcelor Gijon | 2.16 | 1.87 | 0.71 (0.65) | 1.44 | -0.15 | 0.77 (0.72) | 1.42 | -0.01 | 0.76 (0.74) |
| Arcelor Dunkerque | 1.62 | 1.09 | 0.74 (0.63) | 1.33 | 0.04 | 0.74 (0.71) | 1.32 | 0.03 | 0.74 (0.72) |
| Arcelor FOS | 2.51 | 1.93 | 0.73 (0.68) | 1.64 | -0.07 | 0.78 (0.80) | 1.63 | 0.01 | 0.78 (0.81) |
| Port Talbot S Works | 1.98 | 1.83 | 0.79 (0.65) | 1.00 | -0.10 | 0.82 (0.76) | 1.00 | 0.00 | 0.82 (0.78) |
| Scunthorpe IS Works | 2.16 | 2.11 | 0.78 (0.78) | 0.97 | -0.03 | 0.80 (0.78) | 0.97 | 0.01 | 0.80 (0.79) |
| ISD Dunaferr | 3.51 | 3.49 | 0.77 (0.82) | 0.91 | 0.01 | 0.84 (0.82) | 0.91 | 0.00 | 0.84 (0.82) |
| ILVA Taranto | 2.86 | 2.28 | 0.59 (0.57) | 1.37 | -0.19 | 0.71 (0.64) | 1.33 | -0.01 | 0.71 (0.68) |
| TATA IJmuiden | 1.99 | 1.84 | 0.79 (0.86) | 1.20 | 0.01 | 0.78 (0.86) | 1.20 | 0.01 | 0.78 (0.86) |
| Arcelor Dubrowie | 2.82 | 2.77 | 0.75 (0.75) | 1.06 | -0.07 | 0.79 (0.76) | 1.05 | -0.02 | 0.79 (0.77) |
| Arcelor Galati | 3.27 | 3.26 | 0.69 (0.62) | 0.97 | -0.09 | 0.71 (0.65) | 0.96 | 0.01 | 0.71 (0.68) |
| U.S. Steel s.r.o. | 3.45 | 3.41 | 0.77 (0.73) | 1.22 | -0.20 | 0.82 (0.78) | 1.20 | 0.01 | 0.81 (0.77) |

**Table D1.** Comparison between TROPOMI observations and the model for the different locations using the prior estimates (left), only correcting the background (middle), and the posterior estimates (right). The subcolumns show the mean absolute error, the mean error and the correlation between simulation and TROPOMI observation. The values between brackets represent the correlations within 0.25° of the plants to focus on the effect of scaling the plants' emission rates. All errors are shown in ppb. Arcelor Ostrava and Provozovna Třinec share the same simulation, as do Hüttenwerke Duisburg and Thyssen Schwelgern.



**Appendix E: Posterior estimates**

Figure E1 shows the same data as Figure 5 in a scatter plot with the same color scheme. The lines show linear regressions between posterior and reported emissions for the different reporting methods (measured, calculated, and estimated) and for the entire set of plants. The full comparison shows a high correlation 0.86 and a slope of 0.83. The slope smaller than 1 reflects that the TROPOMI-based emission estimates are lower than those reported by the facilities for most plants. Of the different subsets, the reported emissions based on estimation show the biggest deviation from 1 in their slope although the correlation is

high due to the very small number of data points.

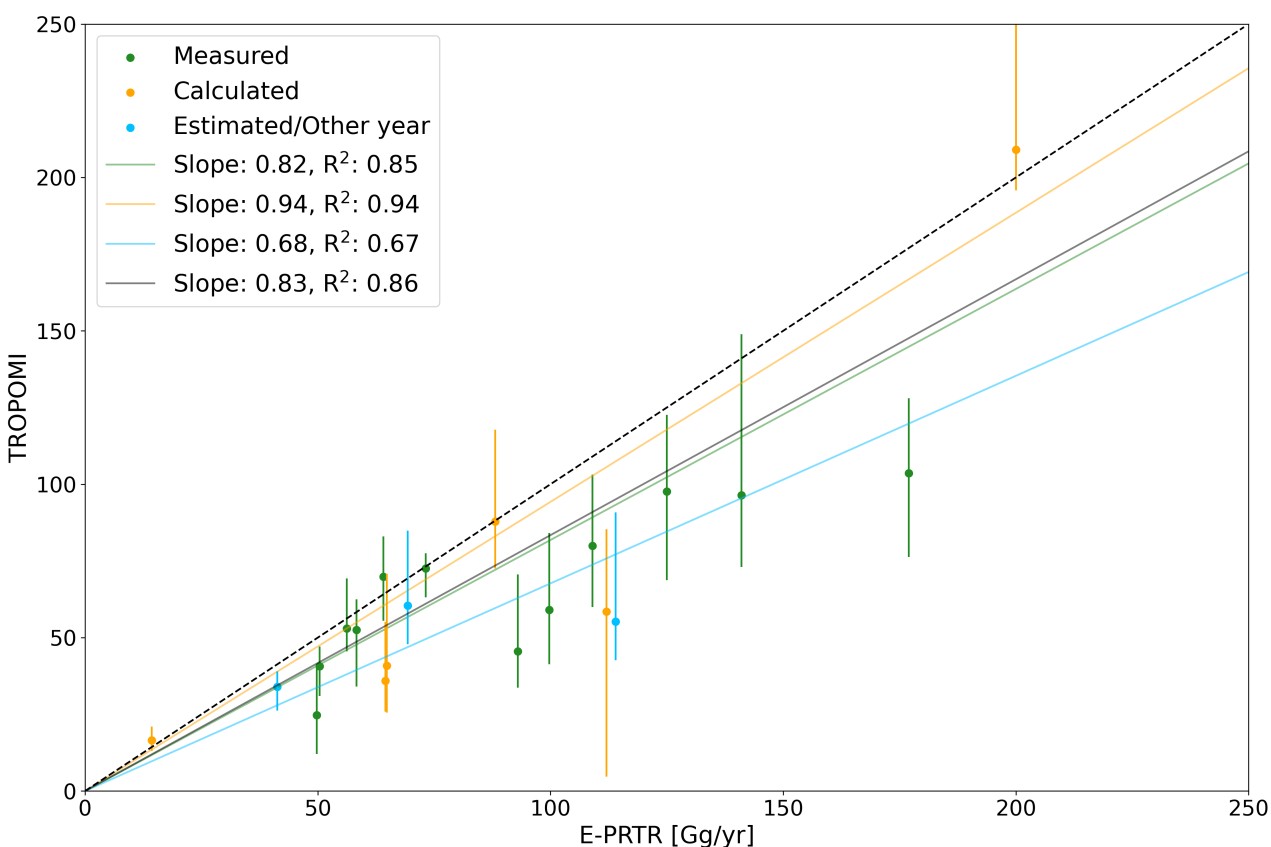

**Figure E1.** Different representation of the data shown in Figure 5, including the correlation between the datasets. The gray line uses the full set of investigated plants whereas the colored lines correspond to subsets using different E-PRTR reporting techniques.



*Author contributions.* GL and JDM designed the study. JDM provided the initial version of the inversion code, for which the WRF model setup was developed with support from IvdV. GL performed the TROPOMI analysis with contributions from JDM, HACDvdG, AS, and IA. GL and JDM wrote the paper with contributions from all authors. HACDvdG and AS provided the TNO emission inventory and associated support. TB provided the TROPOMI carbon monoxide data and associated support.

*Competing interests.* Ilse Aben is a member of the editorial board of ACP.

*Acknowledgements.* We thank the team that realized the TROPOMI instrument and its data products, consisting of the partnership between Airbus Defence and Space Netherlands, KNMI, SRON and TNO, commissioned by NSO and ESA. Sentinel-5 Precursor is part of the EU Copernicus program, and Copernicus Sentinel-5P data (2019-2020) have been used. Part of this work was carried out on the Dutch national e-infrastructure, and we thank SURF (www.surf.nl) for the support in using the National Supercomputer Snellius. The present work was partly
funded through the CORSO project which received funding from European Union's Horizon 2020 Research and Innovation Programme under grant agreement No101082914. TB acknowledges funding by the NSO TROPOMI national program.



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
