# Peer review of "Comparing space-based to reported carbon monoxide emission estimates for Europe's iron & steel plants."

_EGUsphere, 2024_

## Referee Comment (RC1)

This paper describes an analytical inversion to estimate emissions from several plants from the TROPOMI. It also briefly compares with the other two algorithms and find agreement between the inversion results and emissions by CSF method. The combination of the WRF model and the analytical inversion approach for deriving fluxes incurs substantial costs. The amount of work needed to derive fluxes for each industrial source if this is employed is quite computationally expensive. I am not sure this has general applicability. The CSF method is typically utilized. Compared to the CSF method, the combination of the WRF model and the analytical inversion approach does not seem to demonstrate additional advantages. The paper is nicely organized and results summarized well. Some key issues:

(1) I would like to know how many plumes from each of these 21 plants in 2019 can be used to calculate flux.

(2) Clarify the applicable scenarios for the inversion approach, such as beneficial observation conditions.

(3) How is the plume emission height taken into account in all the methods applied in this paper? For example, when using the CSF method, the authors employed the 10-meter wind field from ERA5 to estimate emissions, which might be inaccurate. The emission plumes from steel plants could reach several hundred meters in altitude.

(4) Line 83: I am not sure why the authors are using Data Quality Value (QA Value) below 0.7?

(5) Line 255: Which specific area do the simulated concentrations above the iron and steel plants refer to? It's not clearly indicated in the figure. It would be easier to identify if the TROPOMI observations, the prior and posterior simulation were all displayed.

(6) I suggest that the authors include a geographical distribution map of these steel plants.

(7) How is the annual emission rate for each steel plant determined?

(8) In Section 3.2, the authors mention, "Figure 5 shows the comparison of the posterior emission estimates based on TROPOMI with the prior emission rates from E-PRTR. Nine out of 21 posterior emission estimates are consistent with E-PRTR within their uncertainty ranges." However, I see that 10 estimates (highlighted in the red box) in Figure 5 are consistent with E-PRTR within their uncertainty ranges, not nine.

[Figure]

(9) In Section 3.2, the authors also mention that 10 posterior estimates are within 20% of the reported values. This result is not easily discernible from Figure 5. I suggest marking these 10 steel plants, as well as the 9 steel plants mentioned in the previous comment, separately.

---

## Author Comment (AC1)

We thank both reviewers for their comments. We will reply to each comment individually in the document below, where we have marked text from the reviewers in blue, and alterations to the manuscript in **bold**.

**Review 1:**

This paper describes an analytical inversion to estimate emissions from several plants from the TROPOMI. It also briefly compares with the other two algorithms and find agreement between the inversion results and emissions by CSF method. The combination of the WRF model and the analytical inversion approach for deriving fluxes incurs substantial costs. The amount of work needed to derive fluxes for each industrial source if this is employed is quite computationally expensive. I am not sure this has general applicability. The CSF method is typically utilized. Compared to the CSF method, the combination of the WRF model and the analytical inversion approach does not seem to demonstrate additional advantages. The paper is nicely organized and results summarized well. Some key issues:

We thank the reviewer for the assessment. We have modified the text to emphasize advantages of the inversion approach over mass-balance methods such as the CSF. In addition we would like to refer to line 328-331, which we have expanded upon.

(Line 150-154) We use an analytical inversion to estimate posterior emissions as described in Brasseur and Jacob (2017). **An advantage of the inversion framework over mass-balance approaches is that it more precisely resolves transport in the emission estimation. This comes at the cost of a higher computational load, but improves the accuracy of the emission estimate, and allows the inversion method to be applied to low-coverage situations which would pose challenges to mass-balance methods. In addition, the inclusion of more data allows for estimation of smaller emissions over longer time periods.**

(Line 328-331) Seven of the studied plants; Arcelor Dunkerque, Gent, Dubrowie, Port Talbot S Works, Thyssen Schwelgern, Hüttenwerke Duisburg, and U.S. Steel s.r.o., have reported emission rates above 100 Gg per year and therefore merit the application of the CSF. **The other plants have emission rates lower than what Leguijt et al. (2023) report as the lower limit for the CSF to be reliable.**

(1) I would like to know how many plumes from each of these 21 plants in 2019 can be used to calculate flux.

We have added a line containing the number of days contributing to the inversion emission estimate as well as an explanation of the inclusion of days without clear visible plumes. In addition, we have added a line with the number of orbits on which the CSF was applied. The CSF is also applied to days without a clearly observable plume, to avoid a sampling bias when calculating annual emission rates, as in (Leguijt, G., Maasakkers, J. D., Denier van der Gon, H. A., Segers, A. J., Borsdorff, T., and Aben, I.: Quantification of carbon monoxide emissions from African cities using TROPOMI, Atmospheric Chemistry and Physics Discussions, pp. 1–27, 2023).

(Line 199-203) To further limit the contribution of **spatial** concentration-mismatches we remove days which have the 20% highest optimized observational cost normalized by the number of pixels. **This removes days on which, after aggregation, none of the simulated plumes spatially matched the TROPOMI observed plume well. To ensure representative sampling, we use the optimized observational cost instead of the prior observational cost to avoid structural removal of days without clear observed plume signals.**

(Line 293-295) Most of the plants, 67% (90%), have inversion averaging kernel values above 0.8 (0.7) respectively. **This is a result of the large number of TROPOMI observations, with each plant having TROPOMI measurements covering at least part of the simulated plume on 150 to 250 days.**

(Line 332-334) Figure 6 shows a comparison between the two methods for the applicable plants. **The annual CSF estimates show the mean of the estimates on individual orbits, where the number of suitable orbits ranges from 120 (Port Talbot S Works) to 220 (Arcelor Gent).**

(2) Clarify the applicable scenarios for the inversion approach, such as beneficial observation conditions.

We have expanded upon the applicability of the inversion approach and its advantages in low-coverage situations. We have also changed line 203-206, to emphasize that we are combining all orbits into a single inversion.

(Line 150-153) We use an analytical inversion to estimate posterior emissions as described in Brasseur and Jacob (2017). **An advantage of the inversion framework over mass-balance approaches is the explicit inclusion of transport in the emission estimation. This comes at the cost of a higher computational load, but improves the accuracy of the emission estimate, and allows the inversion method to be applied to low-coverage situations which would pose challenges to mass-balance methods.**

(Line 203-206) The daily inversions are only used for selection of the best simulation on each day. Afterwards, the best daily simulations are combined into **a single Jacobian and prior vector, and then used in** an inversion spanning the full year to determine **an** annual **scaling factor for each emission element of the state vector (x). The emission estimates for the iron & steel plants can therefore be directly compared with the annual emission rates reported to E-PRTR.**

(3) How is the plume emission height taken into account in all the methods applied in this paper? For example, when using the CSF method, the authors employed the 10-meter wind field from ERA5 to estimate emissions, which might be inaccurate. The emission plumes from steel plants could reach several hundred meters in altitude.

Injection height is indeed important to take into account in both the inversion and CSF approach. We have clarified our approach.

(Line 126-128) Both the E-PRTR and TNO emissions are put on a three-dimensional grid using the sector specific vertical profiles provided by Bieser et al. (2011). **The vertical profiles account both for the injection height and include an effective plume rise parametrization.**

(Line 252-255) We perform the CSF as in Leguijt et al. (2023) using **the effective wind calibration and** 10-meter altitude winds from ERA5 (Hersbach et al., 2020). **The effective wind speed is a parametrization of the actual wind speed, which aims to account for the effects of turbulence, variation in the vertical wind profile, and plume rise.**

(4) Line 83: I am not sure why the authors are using Data Quality Value (QA Value) below 0.7?

We have rephrased line 83 to make it clearer that we are not using TROPOMI observations with this QA Value.

(Line 83-84) We only use observations with sensitivity to concentrations at the surface and therefore **remove** observations with a TROPOMI Data Quality Value (QA Value) below 0.7.

(5) Line 255: Which specific area do the simulated concentrations above the iron and steel plants refer to? It's not clearly indicated in the figure. It would be easier to identify if the TROPOMI observations, the prior and posterior simulation were all displayed.

We have modified the figure, which now indicates the region mentioned.

(Line 278-280) Although the largest part of the domain shows better agreement to the observations than the prior simulation (mean bias: -0.11, mean absolute bias: 0.97, correlation: 0.83), the simulated concentrations above the iron & steel plant, **indicated with the black square,** show significant differences.

(Caption figure 4) **The black squares, with dimensions of 0.15◦x 0.15◦, are centered on the location of the plant.**

[Figure]

(6) I suggest that the authors include a geographical distribution map of these steel plants.

We have added a map that shows the locations of the studied plants to Appendix A.

[Figure]

(7) How is the annual emission rate for each steel plant determined?

We have modified line 204, to clarify that we are performing an annual inversion rather than multiple inversions on individual days. This means that all the observations throughout the year are combined into observation vector $y$ in equation 1.

(Line 203-206) The daily inversions are only used for selection of the best simulation on each day. Afterwards, the best daily simulations are combined into **a single Jacobian and prior vector, and then used in** an inversion spanning the full year to determine **an** annual emission rate estimate **for each element of the state vector (x). The emission estimates for the iron & steel plants can be directly compared with the annual emission rates reported to E-PRTR.**

(8) In Section 3.2, the authors mention, "Figure 5 shows the comparison of the posterior emission estimates based on TROPOMI with the prior emission rates from E-PRTR. Nine out of 21 posterior emission estimates are consistent with E-PRTR within their uncertainty ranges." However, I see that 10 estimates (highlighted in the red box) in Figure 5 are consistent with E-PRTR within their uncertainty ranges, not nine.

[Figure]

We thank the reviewer for the attentiveness. There are indeed 10 plants for which the estimates agree within uncertainties. We have corrected the text.

(Line 287-288) Figure 5 shows the TROPOMI-based posterior emission estimates compared to the prior emission rates from E-PRTR. **Ten** out of twenty-one posterior emission estimates agree with E-PRTR within their uncertainty range.

(Line 381-383) Our emission estimates show a high correlation of 0.86 to the emission rates reported to E-PRTR with **10** out of 21 plants agreeing within estimated uncertainties.

(9) In Section 3.2, the authors also mention that 10 posterior estimates are within 20% of the reported values. This result is not easily discernible from Figure 5. I suggest marking these 10 steel plants, as well as the 9 steel plants mentioned in the previous comment, separately.

With the number of plants agreeing within uncertainty being 10, rather than 9, we have decided to remove the remark about the 10 plants agreeing within 20%. Although the statement is valid, there is a lot of overlap between the two sets of plants, and we feel the two different designations would cause confusion.

**Review 2:**

Leguijt et al. present a comprehensive analysis of carbon monoxide emissions from iron and steel plants over Europe. They performed both analytical inversions and cross-sectoral flux estimates using TROPOMI observations, which are further compared against the facility-level reports. They have shown reasonably good agreement in the flux estimates across different approaches, considering error bars. They also investigated if TROPOMI could detect changes in emissions between years.
Overall, the manuscript is well written and most of the results are straightforward. The authors did a great job of providing background about the iron and steel industry and the different approaches used. I enjoy reading this paper and thereby recommend its publication after providing clarifications for a few minor comments:

We thank the reviewer for the kind words.

P1L14 –regarding year-to-year variability: what about temporal variability across different months or overpasses? Can the authors resolve or constrain the emissions at a finer temporal resolution beyond one year? Also, see below comments relating to P8L186-188.

Although we would like to resolve emissions at higher temporal resolution, the degrees of freedom for some of the plants (especially the ones with fewer usable observations or lower emission rates) are not sufficient to allow shorter time-periods. To treat all plants equally, we decided to limit ourselves to yearly estimates. We have added this to the manuscript as a clarification. Moreover, the reporting is only available by year, so we would not be able to evaluate against reported values, which is one of the main objectives.

(Line 298-300) **With 33% of the plants having inversion averaging kernel values below 0.8, we are limited to annual emission rate estimates. 6-monthly emission estimates for these plants would result in averaging kernel values as low as 0.45, and, consequently, emission estimates that are strongly driven by the prior value.**

P3L87-88: What prevents the authors from extending the flux estimates for 2020 for all iron and steel production? Lack of observed data or reported data from E-PRTR? Because only 4 of the total plants see substantial changes in emissions between years?

We are not limited by data, but rather by computation time and we have therefore focused on interesting cases. We have added a line to reflect this limitation.

(Line 86-90) For all iron & steel plants we analyse TROPOMI data for 2019. **In addition,** we analyse 2020 data for four plants, Arcelor Gent (Belgium), Gijon (Spain), Ostrava (Czech Republic) and Provozovna Trinec (Czech Republic). **Because of the heavy computational footprint of our analysis, we limit our 2020 analysis to these plants, which show interesting results for 2019 that warrant further investigation.**

P4L96-100: I may miss this, but does the E-PRTR provide uncertainty estimates to either their measurement or calculation as well, e.g., in Fig. 5?

We have added a line about the absence of reported uncertainty estimates.

(Caption Figure 1) The year 2019 (middle), which is used for the analysis in this work is shown more opaquely. **E-PRTR does not provide uncertainty estimates.**

P6: When inverting emissions using WRF, would there be any nearby emission sources apart from the individual iron and steel plants? How would the authors deal with those non-iron and steel emission sources, e.g., as in the background? In other words, were most of the plants relatively isolated?

We have added a line about the contribution of the iron & steel plants to total local emissions. In addition, the surrounding emissions are part of state-vector elements with contributions throughout the domain (e.g. road transport) and are therefore constrained by observations in the full domain.

> (Line 114-117) As we use the iron & steel plant emission rate from E-PRTR, we remove the corresponding point sources from the TNO GHGco inventory to avoid double counting of emissions. **Because of the CO-intense processes taking place in the iron & steel plants, the emission rates reported to E-PRTR comprise at least 70% of the total emissions in a 0.4°x0.4° box centered on the plant.**

P6L123: How have injection height and plume height been considered in WRF simulations? I suppose those are taken care of by the "sector-specific vertical profiles"?

This is indeed the case, and we have added this to the manuscript as a clarification.

> (Line 126-128) Both the E-PRTR and TNO emissions are put on a three-dimensional grid using the sector specific vertical profiles provided by Bieser et al. (2011). **The vertical profiles account both for the injection height and include an effective plume rise parametrization.**

P7:163- 165: The choice of uncertainties seems a bit arbitrary (e.g., 20% for prior, 50% for transport? etc), even though the authors later performed a set of sensitivity tests of the impact of uncertainties on flux estimates. I would suggest at least providing a few more references or reasoning to those numbers.

We have added our reasoning and a reference for our choice of uncertainty for the TNO inventory and an additional reference for the CAMS uncertainty.

> (Line 174-179) To construct $S_A$, we assume a diagonal shape and an uncertainty of 20% for the TNO GHGco inventory, **in accordance with the 2-sigma range of 38% given in Super et al. (2020)**. We choose an uncertainty of 10% for the CAMS background following e.g. Maasakkers et al. (2022b)**; Naus et al. (2023)**, and a 50% uncertainty on the 4 elements adjusting for inflow from the outer domain reflecting the high uncertainties associated with long-range transport. **As these state vector elements affect many observations, they tend to be well constrained by the observations and changing their prior uncertainty has limited effect on the outcome of the optimization.**

P7L174: It seems that the spatial mismatches between modeled and observed plumes can likely be minimized by a plume-rotation algorithm, while the authors rely on an ensemble of model plumes (Fig.2 right) for "best" alignment. What if none of the ensembles match the observed plumes perfectly (e.g., a near-field bias in wind direction, esp when the plume is curving)?

We have expanded upon the early text to make it clear how we handle days with a residual mismatch between observation and simulation. The removal of "worst matches" is also part of the uncertainty ensemble. While plume rotation is an interesting suggestion, and one that is often applied in mass-balance techniques, it is harder to implement in an inverse framework. One would have to consistently adjust the other modeled tracers as well. It could also lead to an overestimation for small sources, as the rotation may be optimized towards positive noise. We have therefore not included this concept in our manuscript.

> (Line 199-201) To further limit the contribution of **spatial** concentration-mismatches we remove days which have the 20% highest optimized observational cost normalized by the number of pixels. **This removes days on which, after aggregation, none of the simulated plumes spatially matched the TROPOMI observed plume well.**

P8L186-188: I am a bit confused by the aggregation of inversion results here. For example, how many days of TROPOMI have been combined to optimize an annual flux rate? Were the authors able to resolve for a posteriori that is spatially resolved (e.g., posterior flux rate per emission grid) or just one number per plant per year? Were there any possible sampling biases across seasons given cloud interference? I would suggest the authors provide more info on the number of

TROPOMI overpasses being examined and whether those overpasses are representative of an annual average (if they have not done that).

We have optimized for each state vector element, which means an annual scaling per sector and daily scaling of the background. We have added this to the manuscript. Additionally, we have added information about the number of observation days going into the inversion and their temporal distribution.

> (Line 158-160) **The elements of x correspond to annual emissions from the iron & steel plant, the domain-wide emissions from individual sectors in the TNO GHGco inventory, and emissions from the four quadrants of the outer domain as well as the CAMS-based background concentrations.**

> (Line 203-206) Afterwards, the best daily simulations are combined into **a single Jacobian and prior vector, and then used in** an inversion spanning the full year to determine **an** annual emission rate estimate **for each element of the state vector (x). The emission estimates for the iron & steel plants can be directly compared with the annual emission rates reported to E-PRTR.**

> (Line 293-295) Most of the plants, 67% (90%), have inversion averaging kernel values above 0.8 (0.7) respectively. **This is a result of the large number of TROPOMI observations, with each plant having TROPOMI measurements covering at least part of the simulated plume on 150 to 250 days.**

> (Line 280-285) This shows the inversion framework is sensitive to emissions from the iron & steel plant specifically. The corresponding values for the other plants are shown in Appendix D. **To test whether there is a temporal sampling bias in our method, we investigate the distribution of TROPOMI observations throughout the year per plant. The number of observations shows limited variation over the four quarters of the year, ranging from 19.2% of the annual number of valid observations (4th quarter, Arcelor Bremen) to 29.6% (3rd quarter, Scunthorpe IS Works).**

> These lines also relate to the other comment – could the authors try to resolve the emissions from individual plants at a finer resolution beyond just one year?

We have added lines to the manuscript which mention that we are limited to annual scaling.

> (Line 203-206) The daily inversions are only used for selection of the best simulation on each day. Afterwards, the best daily simulations are combined into **a single Jacobian and prior vector, and then used in** an inversion spanning the full year to determine **an** annual emission rate estimate **for each element of the state vector (x). The emission estimates for the iron & steel plants can be directly compared with the annual emission rates reported to E-PRTR.**

> (Line 298-300) **With 33% of the plants having inversion averaging kernel values below 0.8, we are limited to annual emission rate estimates. 6-monthly emission estimates for these plants would result in averaging kernel values as low as 0.45, and, consequently, emission estimates that are strongly driven by the prior value.**

> P9L196-197: Interesting - I was especially intrigued by the co-assimilation of background values with the fluxes! Could the authors provide more info or reference on such inversion construction? For example, any error correlation between the background (mean + gradient) and the fluxes? How much adjustment was made to background vs. plume signals using TROPOMI observations? Relating to the 10% error in background assumed on L163 – what would be prior errors for background mean and background gradients? Providing some supplementary details on the prior error and the Jacobian, particularly for the background optimization would be very helpful for readers.

We have added a reference to a previous paper which uses daily background optimization. In addition, we mention explicitly that both state-vector elements inherit the CAMS-assigned uncertainty. When setting a

higher uncertainty (30%) on the 'background gradients', the resulting change in the estimated annual plant emission rates is less than 1%, as the background tracers are constrained by every observation in the domain. Adjustments to the background are small (0-3% for the mean background, typical adjustments to the gradient are also of the order of 0-3%, with outliers of up to 30% reduction). We have added this to the manuscript. Error correlations between the daily mean background and annual plume optimizations range from 0.005 to 0.15, with the majority of the days centered around 0.05, error correlations between the plume and background gradients are an order of magnitude lower.

(Line 210-212) To reduce the impact of mismatches between the simulated and observed background, we allow our inversion to optimize the background at daily rather than yearly frequency to prevent biases from aliasing into the emissions estimate **(Naus et al., 2023)**.

(Line 214-221) These two parts of the background are added individually to the state vector, yielding two state vector elements per overpass of TROPOMI and giving additional flexibility to the inversion. Panel 2F-I and 2N-O show this flexibility results in a reduced spatial gradient in the posterior simulations, better matching the TROPOMI observation. **Being derived from CAMS, both the mean daily background, and the deviation from the mean are given a 10% uncertainty. Like the state vector elements for transport from the outer domain, the background is well constrained by the large number of TROPOMI observations, resulting in limited sensitivity to the imposed prior uncertainty. Typical adjustments to both the mean background concentration and its gradient range from 0-3%. However, the daily background gradient state vector element gets reduced by up to 30% to better match observations.**

P12L263-268: Glad that the authors also reported the averaging kernel (Eq.5) of their inversion. Very minor point - I would probably differentiate the word choices since TROPOMI also has its own "averaging kernel" from the retrieval.

Throughout the manuscript, we have changed "averaging kernel" to either "averaging kernel of the TROPOMI retrieval" or "averaging kernel of our inversion / inversion averaging kernel" to avoid confusion.

Sect. 3.4 – the 2020 analysis: I am slightly confused by these comparisons and their implications. What drives the smaller year-to-year changes in TROPOMI-constraint emissions compared to E-PRTR reports? Were the authors implying that the wind directional biases may be the driver? Does TROPOMI sampling differ greatly between years as well?

This confusion might be partially explained by us not being clear enough about our intentions, which we aimed to improve in our answer to the next point by the reviewer. The 2019 and 2020 inversions have both different prior estimates (yearly reports from E-PRTR) and different observations (daily observations from the TROPOMI satellite from different years). The posterior estimates for all inversions are mainly driven by the observations, as reflected by the high inversion averaging kernel elements. For Gent and Gijon, we find that the noticeable drop in reported emissions is not reflected in the posterior estimate.
We perform the wind rotation analysis as an alternative simple approach to verify that the TROPOMI-measured concentrations are indeed following the trends of our posterior estimates (Line 360). Only for Ostrava and Trinec, we find that the winds used in the rotation may not be as well aligned with the plume in both years, making the rotated concentration fields difficult to interpret.
To further check whether the inversions were performing similarly in both years, we added a line on correlation between posterior and observation.
As for sampling differences, the 2020 simulations lay within the numbers quoted for number of days with observation (Line 293-295) and differences in quarterly coverage (Line 280-285).

(Line 357-359) **Correlations between simulation and observation are similar between 2020 and 2019; an average posterior (prior) correlation of 0.79 (0.74) in 2020 compared to 0.77 (0.75) in 2019, indicating comparable inversion performances.**

(Line 293-295) Most of the plants, 67% (90%), have inversion averaging kernel values above 0.8 (0.7) respectively. **This is a result of the large number of TROPOMI observations, with each**

**plant having TROPOMI measurements covering at least part of the simulated plume on 150 to 250 days.**

(Line 280-285) This shows the inversion framework is sensitive to emissions from the iron & steel plant specifically. The corresponding values for the other plants are shown in Appendix D. **To test whether there is a temporal sampling bias in our method, we investigate the distribution of TROPOMI observations throughout the year per plant. The number of observations shows limited variation over the four quarters of the year, ranging from 19.2% of the annual number of valid observations (4th quarter, Arcelor Bremen) to 29.6% (3rd quarter, Scunthorpe IS Works).**

P18L340: A more general, clarification question – Do the authors "trust" more of the report from E-PRTR or the inversion results from TROPOMI (yet, E-PRTR and TNO inventory are used as priors)? Were E-PRTR reports in turn used as a dataset to validate the TROPOMI-based inversion (e.g., many figures in the results)? Or both the reports and TROPOMI-based posteriors are not treated as "truth"?

In general, we expect the reporting program data to be reliable and serving as a way of evaluating our inversion results. However, we do find outliers in our analysis, where the satellite data identify potential uncertainties in the reported figures. To clarify our two-way goal, we have rewritten the first part of our conclusion.

(Line 373-379) We performed analytical inversions with 2019 TROPOMI satellite data to determine annual carbon monoxide emission rates for 21 European integrated iron & steel plants. These plants are the highest emitting CO point-sources in Europe. **We compared our top-down emission rate estimates to bottom-up emission rates reported to E-PRTR at facility-level. In doing this, we evaluated limitations of the satellite-based approach, but also identified outliers pointing at uncertainties in the reported data. The E-PRTR emission rates** are used as prior estimates in our inversions. Per site, the inversion uses one of 8 simulations with different meteorology for each day to reach optimal spatial agreement between observation and simulation. We allow further freedom in the inversion by optimizing the CAMS-based background on a daily basis, rather than performing an annual scaling.